# CONTINUOUS INVARIANCE LEARNING

**Yong Lin** [1*], **Fan Zhou** [2*], **Lu Tan** [4], **Lintao Ma** [2], **Jiameng Liu** [1], **Yansu He** [5], **Yuan Yuan** [6,7],
**Yu Liu** [2], **James Zhang** [2], **Yujiu Yang** [4], **Hao Wang** [3]
[1]The Hong Kong University of Science and Technology, [2]Ant Group, [3]Rutgers University,
[4]Tsinghua University, [5]Chinese University of Hong Kong, [6]MIT CSAIL, [7]Boston College

## ABSTRACT

Invariance learning methods aim to learn invariant features in the hope that they generalize under distributional shifts. Although many tasks are naturally characterized by continuous domains, current invariance learning techniques generally assume categorically indexed domains. For example, auto-scaling in cloud computing often needs a CPU utilization prediction model that generalizes across different times (e.g., time of a day and date of a year), where 'time' is a continuous domain index. In this paper, we start by theoretically showing that existing invariance learning methods can fail for continuous domain problems. Specifically, the naive solution of splitting continuous domains into discrete ones ignores the underlying relationship among domains, and therefore potentially leads to suboptimal performance. To address this challenge, we propose Continuous Invariance Learning (CIL), which extracts invariant features across continuously indexed domains. CIL is a novel adversarial procedure that measures and controls the conditional independence between the labels and continuous domain indices given the extracted features. Our theoretical analysis demonstrates the superiority of CIL over existing invariance learning methods. Empirical results on both synthetic and real-world datasets (including data collected from production systems) show that CIL consistently outperforms strong baselines among all the tasks.

## 1 INTRODUCTION

Machine learning models have shown impressive success in many applications, including computer vision (He et al., 2016), natural language processing (Liu et al., 2019), speech recognition (Deng et al., 2013), etc. These models normally take the independent identically distributed (IID) assumption and assume that the testing samples are drawn from the same distribution as the training samples. However, these assumptions can be easily violated when there are distributional shifts in the testing datasets. This is also known as the out-of-distribution (OOD) generalization problem.

Learning invariant features that remain stable under distributional shifts is a promising research area to address OOD problems. However, current invariance methods mandate the division of datasets into discrete categorical domains, which is inconsistent with many real-world tasks that are naturally continuous. For example, in cloud computing, Auto-scaling (a technique that dynamically adjusts computing resources to better match the varying demands) often needs a CPU utilization prediction model that generalizes across different times (e.g., time of a day and date of a year), where 'time' is a *continuous* domain index. Figure 1 illustrates the problems with discrete and continuous domains.

Consider invariant features that can be generalized under distributional shift and spurious features that are unstable. Let $x$ denote the input, $y$ denote the output, $t$ denote the domain index, and $\mathbb{E}^t$ denote taking expectation in domain $t$. Consider the model composed of a featurizer $\Phi$ and a classifier $w$ (Arjovsky et al., 2019). Invariant Risk Minimization (IRM) proposes to learn $\Phi(x)$ by merely using invariant features, which yields the same conditional distribution $\mathbb{P}^t(y|\Phi(x))$ across all domains $t$, i.e., $\mathbb{P}(y|\Phi(x)) = \mathbb{P}(y|\Phi(x), t)$, which is equivalent to $y \perp t|\Phi(x)$ (Arjovsky et al., 2019). Existing approximation methods propose to learn a featurizer by matching $\mathbb{E}^t[y|\Phi(x)]$ for different $t$ (See Section 2.1 for details).

---

*These authors contributed equally to this work. Corresponding to Lintao Ma(*lintao.mlt@antgroup.com*).

**A case study on REx**. However, in the continuous environment setting, there are only very limited samples in each environment. So the finite sample estimations $\hat{\mathbb{E}}^t[\boldsymbol{y}|\Phi(\boldsymbol{x})]$ can deviate significantly from the expectation $\mathbb{E}^t[\boldsymbol{y}|\Phi(\boldsymbol{x})]$. In Section 2.2, We conduct a theoretical analysis of REx (Krueger et al., 2021), a popular variant of IRM. Our analysis shows that when there is a large number of domains and limited sample sizes per domain (i.e., in the continuous domain tasks), REx fails to identify invariant features with a constant probability. This is in contrast to the results in discrete domain tasks, where REx can reliably identify invariant features given sufficient samples in each discrete domain.

**The generality of our results**. Other popular approximations of IRM also suffer from this limitation, such as IRMv1 (Arjovsky et al., 2019), REx (Krueger et al., 2021), IB-IRM (Ahuja et al., 2021), IRMx (Chen et al., 2022), IGA (Koyama & Yamaguchi, 2020), BIRM (Lin et al., 2022a), IRM Game (Ahuja et al., 2020), Fisher (Rame et al., 2022) and SparseIRM (Zhou et al., 2022). They employ positive losses on each environment $\boldsymbol{t}$ to assess whether $\hat{\mathbb{E}}^t[\boldsymbol{y}|\Phi(\boldsymbol{x})]$ matches with the other environments. However, since the estimations $\hat{\mathbb{E}}^t[\boldsymbol{y}|\Phi(\boldsymbol{x})]$ are inaccurate, these methods fail to identify invariant features. We then empirically verify the ineffectiveness of these popular invariance learning methods in handling continuous domain problems. One potential naive solution is to split continuous domains into discrete ones. However, this splitting scheme ignores the underlying relationship among domains and therefore can lead to sub-optimal performance, which is further empirically verified in Section 2.2

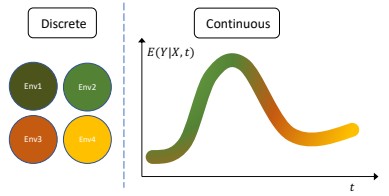

Figure 1: Illustration of distributional shifts in discrete and continuous domains (Wang et al., 2020). Existing IRM methods focus on discrete domains, which is inconsistent with many real-world tasks. Our work therefore aims to extend IRM to continuous domains.

**Our methods**. Recall that our task is to learn a feature extractor $\Phi(\cdot)$ that extracts invariant features from $\boldsymbol{x}$, i.e., learn $\Phi$ which elicits $\boldsymbol{y} \perp \boldsymbol{t}|\Phi(\boldsymbol{x})$. Previous analysis shows that it is infeasible to align $\mathbb{E}^t[\boldsymbol{y}|\Phi(\boldsymbol{x})]$ among different domains $\boldsymbol{t}$ in continuous domain tasks. Instead, we propose to align $\mathbb{E}^y[\boldsymbol{t}|\Phi(\boldsymbol{x})]$ for each class $\boldsymbol{y}$. We can start by training two domain regressors, *i.e.*, $h(\Phi(\boldsymbol{x}))$ and $g(\Phi(\boldsymbol{x}), \boldsymbol{y})$, to regress over the continuous domain index $\boldsymbol{t}$ using L1 or L2 loss. Notably, the L1/L2 distance between the predicted and ground-truth domains captures the underlying relationship between continuous domain indices and can lead to similar domain index prediction losses between domains on both $h$ and $g$ if $\Phi(\boldsymbol{x})$ only extracts invariant features. Specifically, the loss achieved by $h$, i.e., $\mathbb{E}\|\boldsymbol{t} - h(\Phi(\boldsymbol{x}))\|_2^2$, and the loss achieved by $g$, i.e.,

$\mathbb{E}\|\boldsymbol{t} - g(\Phi(\boldsymbol{x}), \boldsymbol{y})\|_2^2$ would be the same if $\Phi(\boldsymbol{x})$ only extracts invariant features. The whole procedure is then formulated into a mini-max framework as shown in Section 3.

In Section 3.1, we emphasize the theoretical advantages of CIL over existing IRM approximation methods in continuous domain tasks. The limited number of samples in each domain makes it challenging to obtain an accurate estimation for $\mathbb{E}^t[\boldsymbol{y}|\Phi(\boldsymbol{x})]$, leading to the ineffectiveness of existing methods. However, CIL does not suffer from this problem because it aims to align $\mathbb{E}^y[\boldsymbol{t}|\Phi(\boldsymbol{x})]$, which can be accurately estimated due to the ample number of samples available in each class (also see Appendix D for the discussion of the relationship between $\mathbb{E}^t[\boldsymbol{y}|\Phi(\boldsymbol{x})]$ and $\mathbb{E}^y[\boldsymbol{t}|\Phi(\boldsymbol{x})]$). In Section 4, we carry out experiments on both synthetic and real-world datasets (including data collected from production systems in Alipay and vision datasets from Wilds-time (Yao et al., 2022)). CIL consistently outperforms existing invariance learning baselines in all the tasks. We summarize our contributions as follows:

- We identify the problem of learning invariant features under continuous domains and theoretically demonstrate that existing methods that treat domains as categorical indexes can fail with constant probability, which is further verified by empirical study.
- We propose Continuous Invariance Learning (CIL) as the first general training framework to address this problem. We also show the theoretical advantages of CIL over existing IRM approximation methods on continuous domain tasks.
- We provide empirical results on both synthetic and real-world tasks, including an industrial application in Alipay and vision datasets in Wilds-time (Yao et al., 2022), showing that CIL consistently achieves significant improvement over state-of-the-art baselines.

## 2 DIFFICULTY OF EXISTING METHODS IN CONTINUOUS DOMAIN TASKS

### 2.1 PRELIMINARIES

**Notations.** Consider the input and output $(\boldsymbol{x}, \boldsymbol{y})$ in the space $\mathcal{X} \times \mathcal{Y}$. Our task is to learn a function $f_\theta \in \mathcal{F} : \mathcal{X} \to \mathcal{Y}$, parameterized by $\theta$. We further denote the domain index as $\boldsymbol{t} \in \mathcal{T}$. We have $(\boldsymbol{x}, \boldsymbol{y}, \boldsymbol{t}) \sim \mathbb{P}(\boldsymbol{x}, \boldsymbol{y}|\boldsymbol{t}) \times \mathbb{P}(\boldsymbol{t})$. Denote the training dataset as $\mathcal{D}^{\text{tr}} := \{(\mathbf{x}_i^{\text{tr}}, \mathbf{y}_i^{\text{tr}}, \mathbf{t}_i^{\text{tr}})\}_{i=1}^n$. The goal of domain generalization is to learn a function $f_\theta$ on $\mathcal{D}^{\text{tr}}$ that can perform well in unseen testing dataset $\mathcal{D}^{\text{te}}$. (This is different from continuously indexed domain adaptation (Wang et al., 2020), as our setting does not have access to target-domain data.) Since the testing domain $\boldsymbol{t}^{\text{te}}$ differs from $\boldsymbol{t}^{\text{tr}}$, $\mathbb{P}(\boldsymbol{x}, \boldsymbol{y}|\boldsymbol{t}^{\text{te}})$ is also different from $\mathbb{P}(\boldsymbol{x}, \boldsymbol{y}|\boldsymbol{t}^{\text{tr}})$ due to distributional shift. Following Arjovsky et al. (2019), we assume that $\boldsymbol{x}$ is generated from invariant feature $\boldsymbol{x}_v$ and spurious feature $\boldsymbol{x}_s$ by some unknown function $\xi$, i.e., $\boldsymbol{x} = \xi(\boldsymbol{x}_v, \boldsymbol{x}_s)$. By the invariance property (more details shown in Appendix C), we have $\boldsymbol{y} \perp \boldsymbol{t}|\boldsymbol{x}_v$ for all $\boldsymbol{t} \in \mathcal{T}$. The target of invariance learning is to make $f_\theta$ only dependent on $\boldsymbol{x}_v$.

**Invariance Learning**. To learn invariant features, Invariant Risk Minimization (IRM) first divides the neural network $\theta$ into two parts, *i.e.*, the feature extractor $\Phi$ and the classifier $\boldsymbol{w}$ (Arjovsky et al., 2019). The goal of invariance learning is to extract $\Phi(\boldsymbol{x})$ which satisfies $\boldsymbol{y} \perp \boldsymbol{t}|\Phi(\boldsymbol{x})$.

**Existing Approximation Methods.** Suppose we have collected the data from a set of discrete domains, $\mathcal{T} = \{\boldsymbol{t}_1, \boldsymbol{t}_2, ..., \boldsymbol{t}_M\}$. The loss in domain $\boldsymbol{t}$ is $\mathcal{R}^t(\boldsymbol{w}, \Phi) := \mathbb{E}^t[\ell(\boldsymbol{w}(\Phi(\boldsymbol{x})), \boldsymbol{y})]$ . Since is hard to validate $\boldsymbol{y} \perp \boldsymbol{t}|\Phi(\boldsymbol{x})$ in practice, existing works propose to align $\mathbb{E}^t[\boldsymbol{y}|\Phi(\boldsymbol{x})]$ for all $t$ as an approximation. Specifically, if $\Phi(\boldsymbol{x})$ merely extract invariant features, $\mathbb{E}^t[\boldsymbol{y}|\Phi(\boldsymbol{x})]$ would be the same in all $t$. Let $\boldsymbol{w}^t$ denote the optimal classifier for domain $\boldsymbol{t}$, i.e., $\boldsymbol{w}^t := \arg\min_{\boldsymbol{w}} \mathcal{R}^t(\boldsymbol{w}, \Phi)$. We have $\boldsymbol{w}^t(\Phi(\boldsymbol{x})) = \mathbb{E}^t[\boldsymbol{y}|\Phi(\boldsymbol{x})]$ hold if the function space of $\boldsymbol{w}$ is sufficiently large (Li et al., 2022). So if $\Phi(\boldsymbol{x})$ relies on invariant features, $\boldsymbol{w}^t$ would be the same in all $\boldsymbol{t}$. Existing approximation methods try to ensure $\boldsymbol{w}^t$ that is the same for all environments to align $\mathbb{E}^t[\boldsymbol{y}|\Phi(\boldsymbol{x})]$ (Arjovsky et al., 2019; Lin et al., 2022a; Ahuja et al., 2021). Further variants also include checking whether $\mathcal{R}^t(\boldsymbol{w}, \Phi)$ is the same (Krueger et al., 2021; Ahuja et al., 2020; Zhou et al., 2022; Chen et al., 2022), or the gradient is the same for all $\boldsymbol{t}$ (Rame et al., 2022; Koyama & Yamaguchi, 2020). For example, REx (Krueger et al., 2021) penalizes the variance of the loss in domains:

$$\mathcal{L}_{\text{REx}} = \sum_{t \in \mathcal{T}} \mathcal{R}^t(\boldsymbol{w}, \Phi) + \lambda|\mathcal{T}|\text{Var}(\mathcal{R}^t(\boldsymbol{w}, \Phi)). \tag{1}$$

where $\text{Var}(\mathcal{R}^t(\boldsymbol{w}, \Phi))$ is the variance of the losses among domains.

**Remark**: The REx loss is conventionally $\mathcal{L}_{\text{REx}} = \frac{1}{|\mathcal{T}|} \sum_{t \in \mathcal{T}} \mathcal{R}^t(\boldsymbol{w}, \Phi) + \lambda\text{Var}(\mathcal{R}^t(\boldsymbol{w}, \Phi))$. We re-scale both terms by $|\mathcal{T}|$ for ease of presentation, which does not change the results.

### 2.2 THE RISKS OF EXISTING METHODS ON CONTINUOUS DOMAIN TASKS

Existing approximation methods propose to learn a feature extractor by matching $\mathbb{E}^t[\boldsymbol{y}|\Phi(\boldsymbol{x})]$ for different $\boldsymbol{t}$. However, in the continuous environment setting, there are a lot of domains and each domain contains limited amount of samples, leading to noisy empirical estimations $\hat{\mathbb{E}}^t[\boldsymbol{y}|\Phi(\boldsymbol{x})]$. Specifically, existing methods employ positive losses on each environment $\boldsymbol{t}$ to assess whether $\hat{\mathbb{E}}^t[\boldsymbol{y}|\Phi(\boldsymbol{x})]$ matches the other environments. However, since the estimated $\hat{\mathbb{E}}^t[\boldsymbol{y}|\Phi(\boldsymbol{x})]$ can deviate significantly from $\mathbb{E}^t[\boldsymbol{y}|\Phi(\boldsymbol{x})]$, these methods fail to identify invariant features. In this part, we first use REx as an example to theoretically illustrate this issue and then show experimental results for other variants.

**Theoretical analysis of REx on continuous domain tasks**. Consider a simplified case where $\boldsymbol{x}$ is a concatenation of a spurious feature $\boldsymbol{x}_s$ and an invariant feature $\boldsymbol{x}_v$, i.e., $\boldsymbol{x} = [x_s, x_v]$. Further, $\Phi$ is a feature mask, i.e., $\Phi \in \{0, 1\}^2$. Let $\Phi_s$ and $\Phi_v$ denote the feature mask that merely select spurious and invariant feature, i.e., $\Phi_s = [1, 0]$ and $\Phi_v = [0, 1]$, respectively. Let $\hat{\mathcal{R}}^t(\boldsymbol{w}, \Phi) = \frac{1}{n^t} \sum_{i=1}^{n_e} \ell(\boldsymbol{w}(\Phi(\boldsymbol{x}_i)), y_i)$ denote the finite sample loss on domain $t$ where $n^t$ is the sample size in domain $t$. Denote $\hat{\mathcal{L}}_{\text{REx}}(\Phi)$ as the finite-sample REx loss in Eq. equation 1. We omit the subscript 'REx' when it is clear from the context in this subsection. REx can identify the invariant feature mask $\Phi_v$ if and only if $\hat{\mathcal{L}}(\Phi_v) < \hat{\mathcal{L}}(\Phi), \forall \Phi \in \{0, 1\}^2$. Suppose the dataset $\mathcal{S}$ that contains $|\mathcal{T}|$ domains. For simplicity, we assume that there are equally $n^t = n/|\mathcal{T}|$ samples in each domain.

**Assumption 1.** *The expected loss of the model using spurious features in each domain, $\mathcal{R}^t(\boldsymbol{w}, \Phi_s)$, follows a Gaussian distribution, i.e., $\mathcal{R}^t(\boldsymbol{w}, \Phi_s) \sim \mathcal{N}(\mathcal{R}(\boldsymbol{w}, \Phi_s), \delta_R)$, where the mean $\mathcal{R}(\boldsymbol{w}, \Phi_s)$ is the average loss over all domains. Given a feature mask $\Phi \in \{\Phi_s, \Phi_v\}$, the loss of an individual sample $(\boldsymbol{x}, y)$ from environment $t$ deviates from the expectation loss (of domain $t$) by a Gaussian, $\ell(y, \boldsymbol{w}(\Phi(\boldsymbol{x}))) - \mathbb{E}^t[\ell(y, \boldsymbol{w}(\Phi(\boldsymbol{x})))] \sim \mathcal{N}(0, \sigma_\Phi^2)$. The variance $\sigma_\Phi$ is drawn from a hyper exponential distribution with density function $p(\sigma_\Phi; \lambda) = \lambda \exp(-\lambda \sigma_\Phi)$ for each $\Phi$.*

**Remark on the Setting and Assumption.** Recall that $\mathcal{R}^t(\boldsymbol{w}, \Phi)$ represents the expected loss of $(\boldsymbol{w}, \Phi)$ in domain $t$. When $\Phi = \Phi_v$, the loss is the same across all domains, resulting in zero penalty. However, when $\Phi = \Phi_s$, the loss in domain $t$ deviates from the average loss, introducing a penalty that increases linearly with the number of environments. In this case, REx can easily identify the invariant feature if we have an infinite number of samples in each domain. However, if we have multiple environments with limited sample sizes in each, REx can fail with a constant probability. This limitation will be discussed further in the following parts.

Let $\mathcal{G}^{-1} : [0, 1] \to [0, \infty)$ denote the inverse of the cumulative density function: $\mathcal{G}(t) = P(z \le t)$ of the distribution whose density is $p(z) = 1 - 1/2 \exp(-\lambda z)$ for $z > 0$. Proposition 1 below shows that REx fails when there are many domains but with a limited number of samples in each domain.

**Proposition 1.** *If $\frac{n}{|\mathcal{T}|} \to \infty$, with probability approaching 1, we have $\mathbb{E}[\hat{\mathcal{L}}(\Phi_v)] < \mathbb{E}[\hat{\mathcal{L}}(\Phi_s)]$, where the expectation is taken over the random draw of the domain and each sample given $\sigma_\Phi$. However, if the domain number $|\mathcal{T}|$ is comparable with $n$, REx can fail with a constant probability. For example, if $|\mathcal{T}| \ge \frac{\sigma_R \sqrt{n}}{\Delta \mathcal{G}^{-1}(1/4)}$, then with probability at least $1/4$, $\mathbb{E}[\hat{\mathcal{L}}(\Phi_v)] > \mathbb{E}[\hat{\mathcal{L}}(\Phi_s)]$.*

**Proof Sketch and Main Motivation.** The complete proof is included in Appendix F. When there are only a few samples in each domain (i.e., $|\mathcal{T}|$ is comparable to $n$), the empirical loss $\hat{\mathcal{R}}^t(\boldsymbol{w}, \Phi)$ deviates from its expectation by a Gaussian variable $\epsilon_t \sim \mathcal{N}(0, \sigma_\Phi^2/n_t)$. After taking square, we have $\mathbb{E}[\epsilon_t^2] = \sigma_\Phi^2/n_t$. There are $|\mathcal{T}|$ domains and $n_t = n/|\mathcal{T}|$. So we have $\sum_t \mathbb{E}[\epsilon_t^2] = |\mathcal{T}|^2 \sigma_\Phi^2/n$, indicating that the data randomness can induce an estimation error that is quadratic in $|\mathcal{T}|$. Note that the expected penalty (assuming we have infinite samples in each domain) grows linearly in $|\mathcal{T}|$. Therefore the estimation error dominates the empirical loss $\hat{\mathcal{L}}$ with large $|\mathcal{T}|$, which means the algorithm selects features based on the data noise rather than the invariance property. Intuitively, existing IRM methods try to find a feature $\Phi$ which aligns $\hat{\mathbb{E}}^t[Y|\Phi(\boldsymbol{x})]$ among different $t$. Due to the finite-sample estimation error, $\hat{\mathbb{E}}^t[Y|\Phi(\boldsymbol{x})]$ can be far away from $\mathbb{E}^t[Y|\Phi(\boldsymbol{x})]$, leading to the failure of invariance learning. One can also show that other variants, e.g., IRMv1, also suffer from this issue (see empirical verification later in this section).

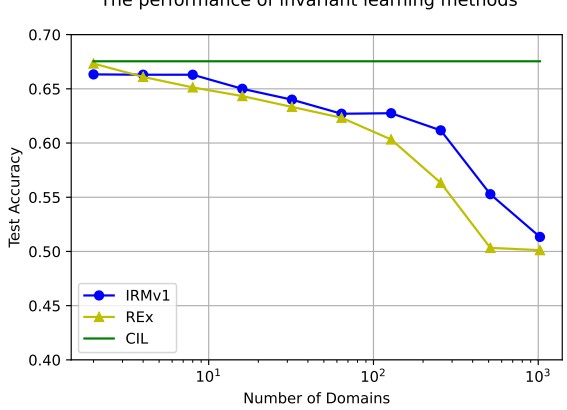

The performance of invariant learning methods

**Implications for Continuous Domains.** Proposition 1 shows that if we have a large number of domains and each domain contains limited data, REx can fail to identify the invariant feature. Specifically, as long as $|\mathcal{T}|$ is larger than $O(\sqrt{n})$, REx can fail even if each domain contains $O(\sqrt{n})$ samples.

When we have a continuous domain in real-world applications, each sample $(\boldsymbol{x}_i, y_i)$ has a distinct domain index, i.e., there is only one sample in each domain. Therefore when $|\mathcal{T}| \approx n$, REx can fail with probability $1/4$ when $n \ge \left(\frac{\sigma_R}{\delta \mathcal{G}^{-1}(1/4)}\right)^2$.

**Empirical verification**. We conduct experiments on CMNIST (Arjovsky et al., 2019) to corroborate our theoretical analysis. The original CMNIST contains 50,000 samples with 2 do-

Figure 2: Empirical validation of how the performance of IRM deteriorates with the number of domains while the total sample size is fixed. The experiments are conducted on CMNIST (Arjovsky et al., 2019) with 50,000 samples. We equally split the original 2 domains into more domains. Since CMNIST only contains 2 classes, 50% test accuracy is close to random guessing. Notably, the data with continuous domains can contain an infinite number of domains with only one sample in each domain.

mains. We keep the total sample size fixed and split the 2 domains into 4, 8, 16,..., and 1024 domains (more details in Appendix L). The results in Figure 2 show that the testing accuracy of REx and IRMv1 decreases as the number of domains increases. Their accuracy eventually drops to 50% (close to random guessing) when the number of domain numbers is 1,024. These results show that existing invariance methods can struggle when there are many domains with limited samples in each domain. Refer Table 1 for results on more existing methods.

| Method | IRMv1 | REx | IB-IRM | IRMx | IGA | InvRat-EC | BIRM | IRM Game | SIRM | CIL (Ours) |
|--------|-------|-----|--------|------|-----|-----------|------|----------|------|------------|
| Acc(%) | 50.4 | 51.7 | 55.3 | 52.4 | 48.7 | 57.3 | 47.2 | 58.3 | 52.2 | **67.2** |

Table 1: OOD performance of existing methods on continuous CMNIST with 1024 domains.

**Further discussion on merging domains**. A potential method to improve existing invariance learning methods in continuous domain problems is to merge the samples with similar domain indices into a 'larger' domain. However, since we may have no prior knowledge of the latent heterogeneity of the spurious feature, merging domains is very hard in practice (also see Section 4 for empirical results on merged domains).

## 3 OUR METHOD

We propose Continuous Invariance Learning (CIL) as a general training framework for learning invariant features among continuous domains.

**Formulation.** Suppose $\Phi(\boldsymbol{x})$ successfully extracts invariant features, and we have $\boldsymbol{y} \perp \boldsymbol{t}|\Phi(\boldsymbol{x})$ according to Arjovsky et al. (2019). The previous analysis shows that it is difficult to align $\mathbb{E}^t[\boldsymbol{y}|\Phi(\boldsymbol{x})]$ for each domain $\boldsymbol{t}$ since there are only very limited samples in $\boldsymbol{t}$ in the continuous domain tasks. This also explains why most existing methods fail in continuous domain tasks. In this part, we propose to align $\mathbb{E}^y[\boldsymbol{t}|\Phi(\boldsymbol{x})]$ for each class $\boldsymbol{y}$ (see Appendix D for the discussion on the relationship between $\mathbb{E}^t[\boldsymbol{y}|\Phi(\boldsymbol{x})]$ and $\mathbb{E}^y[\boldsymbol{t}|\Phi(\boldsymbol{x})]$). Since there is a sufficient number of samples in each class $\boldsymbol{y}$, we can obtain an accurate estimation of $\mathbb{E}^y[\boldsymbol{t}|\Phi(\boldsymbol{x})]$. We perform the following steps to verify whether $\mathbb{E}^y[\boldsymbol{t}|\Phi(\boldsymbol{x})]$ is the same for each $\boldsymbol{y}$ : we first fit a function $h \in \mathcal{H}$ to predict $\boldsymbol{t}$ based on $\Phi(\boldsymbol{x})$. Since $\boldsymbol{t}$ is continuous, we adopt L2 loss to measure the distance between $\boldsymbol{t}$ and $h(\Phi(\boldsymbol{x}))$, i.e., $\mathbb{E}[\|h(\Phi(\boldsymbol{x})) - \boldsymbol{t}\|_2^2]$. We use another function $g$ to predict $\boldsymbol{t}$ based on $\Phi(\boldsymbol{x})$ and $\boldsymbol{y}$, and minimize $\mathbb{E}\|g(\Phi(\boldsymbol{x}), \boldsymbol{y}) - \boldsymbol{t}\|_2^2$. If $\boldsymbol{y} \perp \boldsymbol{t}|\Phi(\boldsymbol{x})$ holds, $\boldsymbol{y}$ does not bring more information to predict $\boldsymbol{t}$ when conditioned on $\Phi(\boldsymbol{x})$. Then the loss achieved by $g(\Phi(\boldsymbol{x}), \boldsymbol{y})$ would be similar to the loss achieved by $h(\Phi(\boldsymbol{x}))$, i.e., $\mathbb{E}[\|g(\Phi(\boldsymbol{x}), \boldsymbol{y}) - \boldsymbol{t}\|_2^2] = \mathbb{E}[\|h(\Phi(\boldsymbol{x})) - \boldsymbol{t}\|_2^2]$.

In conclusion, we solve the following framework:

$$\min_{\Phi, \boldsymbol{w}} \mathbb{E}_{\boldsymbol{x}, \boldsymbol{y}} \ell(\boldsymbol{w}(\Phi(\boldsymbol{x})), \boldsymbol{y}) \quad \text{s.t.} \min_{h \in \mathcal{H}} \max_{g \in \mathcal{G}} \mathbb{E}[\|h(\Phi(\boldsymbol{x})) - \boldsymbol{t}\|_2^2 - \|g(\Phi(\boldsymbol{x}), \boldsymbol{y}) - \boldsymbol{t}\|_2^2] = 0$$

In practice, we can replace the hard constraint in equation 2 with a soft regularization term as follows:

$$\min_{\boldsymbol{w}, \Phi, h} \max_{g} Q(\boldsymbol{w}, \Phi, g, h) = \mathbb{E}_{\boldsymbol{x}, \boldsymbol{y}, \boldsymbol{t}} \Big[ \ell(\boldsymbol{w}(\Phi(\boldsymbol{x})), \boldsymbol{y}) + \lambda(\|h(\Phi(\boldsymbol{x})) - \boldsymbol{t}\|_2^2 - \|g(\Phi(\boldsymbol{x}), \boldsymbol{y}) - \boldsymbol{t}\|_2^2) \Big],$$

where $\lambda \in \mathbb{R}^+$ is the penalty weight.

**Algorithm.** We adapt Stochastic Gradient Ascent and Descent (SGDA) to solve equation 2. SGDA alternates between inner maximization and outer minimization by performing one step of gradient each time. The full algorithm is shown in Appendix A.

**Remark**: Our method is indeed based on the assumption that the class variable is discrete. Invariance learning methods have primarily been applied to classification tasks where the class variable is typically discrete, which aligns with our requirement. It is worth noting that previous methods often assume that the domains are discrete, which may not be applicable in many applications where the domains are continuous.

## 3.1 THEORETICAL ANALYSIS OF CONTINUOUS INVARIANCE LEARNING

In this section, we assume $t$ is a one-dimensional scalar to ease the notation for clarity. All results still hold when $t$ is a vector. We start by stating some assumptions on the capacity of the function classes.

**Assumption 2.** $\mathcal{H}$ contains $h^*$, where $h^*(z) := \mathbb{E}[t|z = \Phi(x)]$.

**Assumption 3.** $\mathcal{G}$ contains $g^*$, where $g^*(z, y) := \mathbb{E}[t|z = \Phi(x), y]$.

Then we have the following results:

**Lemma 1.** *Suppose Assumption 2 and 3 hold, $h^*$ and $g^*$ minimize the following losses given a fixed $\Phi$, we have $h^*(\cdot) = \arg\min_{h \in \mathcal{H}} \mathbb{E}_{(x,t)}(h(\Phi(x)) - t)^2$, and $g^*(\cdot) = \arg\min_{g \in \mathcal{G}} \mathbb{E}_{(x,y,t)}(g(\Phi(x), y) - t)^2$. where the proof is shown in Appendix G.*

**Theorem 2.** *Suppose Assumptions 2 and 3 hold. The constraint of equation 2 is satisfied if and only $\mathbb{E}[t|\Phi(x)] = \mathbb{E}[t|\Phi(x), y]$ holds for $\forall y \in \mathcal{Y}$.*

Proof in Appendix H. The advantage of CIL is discussed as follows:

**The advantage of CIL in continuous domain tasks.** Consider a 10-class classification task with an infinite number of domains and each domain contains only one sample, i.e., $n \to \infty$, and $n/|\mathcal{T}| = 1$, Proposition 1 shows that REx would fail to identify the invariant feature with constant probability. Whereas, Theorem 2 shows that CIL can still effectively extract invariant features. Intuitively, existing methods aim to align $\mathbb{E}^t[y|\Phi(x)]$ across different $t$ values. However, in continuous environment settings with limited samples per domain, the empirical estimations $\hat{\mathbb{E}}^t[y|\Phi(x)]$ become noisy. These estimations deviate significantly from the true $\mathbb{E}^t[y|\Phi(x)]$, rendering existing methods ineffective in identifying invariant features. In contrast, CIL proposes to align $\mathbb{E}^y[t|\Phi(x)]$, which can be accurately estimated as there are sufficient samples in each class.

We have shown that the definite advantage of CIL based on Proposition 1 and Theorem 2. In the following part, we are going to show the finite sample property of our CIL for completeness. This finite sample analysis is a standard analysis of mini-max formulation based on the results of Lei et al. (2021). We consider the empirical counterpart of the soft regularization version (equation 2) for finite sample performance, i.e.,

$$\min_{w, \Phi, h} \max_g \hat{Q}(w, \Phi, g, h) := \hat{\mathbb{E}}_{x,y,t}\Big[\ell(w(\Phi(x)), y) + \lambda\big(\|h(\Phi(x)) - t\|_2^2 - \|g(\Phi(x), y) - t\|_2^2\big)\Big].$$

where $\hat{\mathbb{E}}$ is the empirical counterpart of expectation $\mathbb{E}$. Suppose that we solve equation 2 with SGDA (Algorithm 1) and obtain $(\hat{w}, \hat{\Phi}, \hat{g}, \hat{h})$, which is a $(\epsilon_1, \epsilon_2)$ optimal solution of equation 2. Specifically, we have

$$\hat{Q}(\hat{w}, \hat{\Phi}, \hat{h}, \hat{g}) \le \inf_{w, \Phi, g} Q(w, \Phi, h, \hat{g}) + \epsilon_1, \quad \hat{Q}(\hat{w}, \hat{\Phi}, \hat{h}, \hat{g}) \ge \sup_g Q(\hat{w}, \hat{\Phi}, \hat{h}, g) - \epsilon_2. \quad (2)$$

In the following, we denote $Q^*(w, \Phi, h) := \sup_g Q(w, \Phi, h, g)$. We can see that a small $Q^*(w, \Phi, h)$ indicates that the model $(w, \Phi, h)$ achieves a small prediction loss of $Y$ as well as a small invariance penalty.

**Proposition 2.** *Suppose we solve equation 2 by SGDA as introduced in Algorithm 1 using a training dataset of size $n$ and obtain an $(\epsilon_1, \epsilon_2)$ solution $(\hat{w}, \hat{\Phi}, \hat{g}, \hat{h})$. Under the assumptions specified in the appendix, we then have with probability at least $1 - \delta$ that*

$$Q^*(\hat{w}, \hat{\Phi}, \hat{h}) \le (1 + \eta) \inf_{w, \Phi, h} Q^*(w, \Phi, h) + \frac{1 + \eta}{\eta}\Big(\epsilon_1 + \epsilon_2 + \tilde{O}(1/n)\log(1/\delta)\Big),$$

*where $\tilde{O}$ absorbs logarithmic and constant variables which are specified in the Appendix I.*

**Empirical Verification**. We apply our CIL method on CMNIST (Arjovsky et al., 2019) to validate our theoretical results. We attach a continuous domain index for each sample in CMNIST. The 50,000 samples of CMNIST are simulated to distribute uniformly on the domain index $t$ from 0.0 to 1000.0 (Refer to Appendix J for detailed description). As Figure 2 shows, CIL outperforms REx and IRMv1 significantly when REx and IRMv1 have many domains.

| Env. Type | Method | Linear | | | Sine | | |
|---|---|---|---|---|---|---|---|
| | | Split Num | ID | OOD | Split Num | ID | OOD |
| None | ERM | – | 86.38 (0.19) | 13.52 (0.26) | – | 87.25 (0.46) | 16.05 (1.03) |
| Discrete | IRMv1 | 4 | 51.02 (0.86) | 49.72 (0.86) | 16 | 49.74 (0.62) | 50.01 (0.34) |
| | REx | 8 | 82.05 (0.67) | 49.31 (2.55) | 4 | 81.93 (0.91) | 54.97 (1.71) |
| | GroupDRO | 16 | 99.16 (0.46) | 30.33 (0.30) | 2 | 99.23 (0.04) | 30.20 (0.30) |
| | IIBNet | 8 | 63.25 (19.01) | 38.30 (16.63) | 4 | 61.26 (16.81) | 36.41 (15.87) |
| Continuous | IRMv1 | – | 49.57 (0.33) | 48.70 (2.65) | – | 50.43 (1.23) | 49.63 (15.06) |
| | REx | – | 78.98 (0.32) | 41.87 (0.48) | – | 79.97 (0.79) | 42.24 (0.74) |
| | Diversify | – | 50.03 (0.04) | 50.11 (0.09) | – | 50.07 (0.06) | 50.27 (0.21) |
| | CIL (Ours) | – | 57.35 (6.89) | **57.20 (6.89)** | – | 69.80 (3.95) | **59.50 (8.67)** |

Table 2: Accuracy on Continuous CMNIST for Linear and Sine $p_s(t)$. The standard deviation in brackets is calculated with 5 independent runs. The Env. type "Discrete" means that we manually create by equally splitting the raw continuous domains. The environment type "Continuous" indicates using the original continuous domain index. "Split Num" stands for the number of domains we manually create and we report the best performance among spilt $\{2, 4, 8, 16\}$. Detailed results in Appendix L

# 4 EXPERIMENTS

To evaluate our proposed **CIL** method, we conduct extensive experiments on two synthetic datasets and four real-world datasets, the synthetic logit dataset is presented in Appendix K. we compare CIL with 1) Standard Empirical Risk Minimization (ERM), 2) IRMv1 proposed in Arjovsky et al. (2019), 3) REx in equation 1 proposed in Krueger et al. (2021), and 4) GroupDRO proposed in Sagawa et al. (2019) that minimize the loss of worst group(domains) with increased regularization in training, 5) Diversify proposed in Lu et al. (2022) and 6) IIBNet proposed in Li et al. (2022), adding invariant information bottleneck (IIB) penalty in training. Note that while CIDA (Wang et al., 2020) and its variants (Xu et al., 2022; 2023; Liu et al., 2023) also handle continuously indexed domains, they are domain adaptation methods and therefore not included as baselines (see Appendix B for details). For IRMv1, REx, GroupDRO, and IIBNet, we try them on the original continuous domains as well as manually split the dataset with continuous domains into discrete ones (more details in separate subsections below). All the experiments are repeated at least three times and we report the accuracy with standard deviation on each dataset.

## 4.1 SYNTHETIC DATASETS

### 4.1.1 CONTINUOUS CMNIST

**Setting**. We construct a continuous variant of CMNIST following Arjovsky et al. (2019). The digit is the invariant feature $x_v$ and the color is the spurious feature $x_s$. Our goal is to predict the label of the digit, $y$. We generate 1000 continuous domains. The correlation between $x_v$ and the label $y$ is $p_v = 75\%$, while the spurious correlation $p_s(t)$ changes among domains $t$, whose details are included in the Appendix L. Similar to the previous dataset, we try two settings with $p_s(t)$ being a linear and Sine function.

**Results**. Table 2 reports the training and testing accuracy of methods on CMNIST in two settings. We also tried different domain splitting schemes for IRMv1, REx, GroupDRO, and IIBNet with the complete results in the Appendix L. ERM performs very well in training but worst in testing, which implies ERM tends to rely on spurious features. GroupDRO achieves the highest accuracy in training but the lowest in testing except for ERM. Our proposed CIL outperforms all baselines on two settings by at least 8% and 5%, respectively.

## 4.2 REAL-WORLD DATASETS

### 4.2.1 HOUSEPRICE

We also evaluate different methods on the real-world HousePrice dataset from Kaggle[*]. Each data point contains 17 explanatory variables such as the built year, area of living room, overall condition

---

[*]https://www.kaggle.com/c/house-prices-advanced-regression-techniques

| Env. Type | Method | HousePrice | | Insurance Fraud | | Alipay Auto-scaling | |
|---|---|---|---|---|---|---|---|
| | | ID | OOD | ID | OOD | ID | OOD |
| None | ERM | 82.36 (1.42) | 73.94 (5.04) | 79.98 (1.17) | 72.84 (1.44) | 89.97 (1.35) | 57.38 (0.64) |
| Discrete | IRMv1 | 84.29 (1.04) | 73.46 (1.41) | 75.22 (1.84) | 67.28 (0.64) | 88.31 (0.48) | 66.49 (0.10) |
| | REx | 84.23 (0.63) | 71.30 (1.17) | 78.71 (2.09) | 73.20 (1.65) | 89.90 (1.08) | 65.86 (0.40) |
| | GroupDRO | 85.25 (0.87) | 74.76 (0.98) | 86.32 (0.84) | 71.14 (1.30) | 91.99 (1.20) | 59.65 (0.98) |
| | IIBNet | 52.99 (10.34) | 47.48 (12.60) | 73.73 (22.96) | 69.17 (18.04) | 61.88 (13.01) | 52.97 (12.97) |
| | InvRAT | 83.33 (0.12) | 74.41 (0.43) | 82.06 (0.72) | 73.35 (0.48) | 89.84 (1.38) | 57.54 (0.47) |
| Continuous | IRMv1 | 82.45 (1.27) | 75.40 (0.99) | 54.98 (3.74) | 52.09 (2.05) | 88.57 (2.29) | 66.20 (0.06) |
| | REx | 83.59 (2.01) | 68.82 (0.92) | 78.12 (1.64) | 72.90 (0.46) | 89.94 (1.64) | 63.95 (0.87) |
| | Diversify | 81.14 (0.61) | 70.77 (0.74) | 72.90 (7.39) | 63.14 (5.70) | 80.16 (0.24) | 59.81 (0.09) |
| | IIBNet | 62.29 (4.40) | 53.93 (3.70) | 76.34 (5.20) | 72.01 (6.99) | 80.49 (8.30) | 58.89 (5.53) |
| | InvRAT | 82.29 (0.76) | 77.18 (0.41) | 80.63 (1.04) | 72.07 (0.74) | 88.74 (1.54) | 60.58 (3.22) |
| | EIIL | 82.62 (0.42) | 76.85 (0.44) | 80.60 (1.36) | 72.44 (0.58) | 91.34 (1.50) | 53.14 (0.74) |
| | HRM | 84.67 (0.62) | 77.40 (0.27) | 81.93 (1.11) | 73.52 (0.46) | 89.84 (1.20) | 55.44 (0.35) |
| | ZIN | 84.80 (0.60) | 77.54 (0.30) | 81.93 (0.73) | 73.33 (0.43) | 90.56 (0.91) | 58.99 (0.87) |
| | CIL (L1) | 83.41 (0.75) | 77.98 (1.02) | 82.39 (1.40) | **76.54 (1.03)** | 81.44 (1.77) | 68.51 (1.33) |
| | CIL (L2) | 82.51 (1.96) | **79.29 (0.77)** | 80.30 (2.06) | 75.01 (1.18) | 81.25 (1.65) | **71.29 (0.04)** |

Table 3: Accuracy of each method on three real-world datasets with standard deviation in brackets. Each method takes 5 runs independently. The details of the settings for HousePrice, Insurance Fraud, and Alipay Auto-scaling can be found in Section 4.2.1, 4.2.2, and 4.2.3, respectively. CIL is our method. L1 or L2 means we use the L1 or L2 loss. We adopt L2 loss by default in other tables.

rating, etc. The dataset is partitioned according to the built year, with the training dataset in the period [1900, 1950] and the test dataset in the period (1950, 2000). Our goal is to predict whether the house price is higher than the average selling price in the same year. The built year is regarded as the continuous domain index in CIL. We split the training dataset equally into 5 segments for IRMv1, REx, GroupDRO, and IIBNet with a decade in each segment.

**Results**. The training and testing accuracy is shown in Table 3. GroupDRO performs the best across all baselines both on training and testing, while IIBNet seems unable to learn valid invariant features in this setting. REx achieves high training accuracy but the lowest testing accuracy except for IIBNet, indicating that it learns spurious features. Our CIL outperforms the best baseline by over 5% on testing accuracy, which implies the model trained by CIL relies more on invariant features. Notably, CIL also enjoys a much smaller variance on this dataset.

### 4.2.2 INSURANCE FRAUD

This experiment conducts a binary classification task based on a vehicle insurance fraud detection dataset on Kaggle[*]. After data preprocessing, each insurance claim contains 13 features including demographics, claim details, policy information, etc. The customer's age is taken as the continuous domain index, where the training dataset contains customers with ages between (19, 49) and the testing dataset contains customers with ages between (50, 64). We equally partition the training dataset into discrete domains with 5 years in each domain for existing methods dependent on discrete domains. Results in IRMv1 is inferior to ERM in terms of both training and testing performances. REx only slightly improves the testing accuracy over ERM. Table 3 shows that CIL performs the best across all methods, improving by about 2% compared to other methods.

### 4.2.3 ALIPAY AUTO-SCALING

Auto-scaling (Qian et al., 2022) is an effective tool in elastic cloud services that dynamically scale computing resources (CPU, memory) to closely match the ever-changing computing demand. Auto-scaling first tries to predict the CPU utilization based on the current internet traffic. When the predicted CPU utilization exceeds a threshold, Auto-scaling would add CPU computing resources to ensure a good quality of service (QoS) effectively and economically. In this task, we aim to predict the relationship between CPU utilization and the current running parameters of the server in the Alipay cloud. Each record includes 10 related features, such as the number of containers, and network flow, etc. We construct a binary classification task to predict whether CPU utilization is above the 13% threshold or not, which is used in the cloud resource scheduling to stabilize the cloud system.

---

[*]https://www.kaggle.com/code/girishvutukuri/exercise-insurance-fraud

| Method | ID | OOD |
|---|---|---|
| Fine-tuning | 81.98 | 69.62 |
| EWC (Kirkpatrick et al., 2017) | 80.07 | 66.61 |
| SI (Zenke et al., 2017) | 78.70 | 65.18 |
| A-GEM (Lopez-Paz & Ranzato, 2017) | 81.04 | 67.07 |
| ERM | 79.50 | 63.09 |
| GroupDRO-T (Sagawa et al., 2019) | 77.06 | 60.96 |
| mixup(Zhang et al., 2017a) | 83.65 | 58.70 |
| CORAL-T(Sun & Saenko, 2016) | 77.53 | 68.53 |
| IRM-T (Arjovsky et al., 2019) | 80.46 | 59.34 |
| SimCLR (Chen et al., 2020) | 78.59 | 64.42 |
| Swav (Caron et al., 2020) | 78.38 | 60.15 |
| SWA (Izmailov et al., 2018) | 84.25 | 67.90 |
| **CIL (Ours)** | 82.89 | **71.22** |

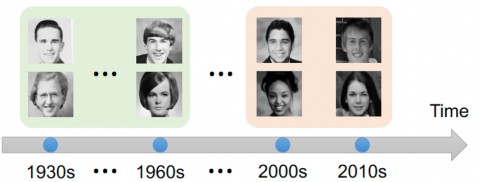

Distribution shift over time

1930s ··· 1960s ··· 2000s 2010s — Time

Figure 3: An illustration Yearbook (Yao et al., 2022). Images taken from Yao et al. (2022).

Figure 4: The accuracy on the worst test OOD domain of each method on Yearbook dataset on Wild-time. The performance of baseline methods is copied from Yao et al. (2022).

We take the minute of the day as the continuous domain index. The dataset contains 1440 domains with 30 samples in each domain. We then split the continuous domains by consecutive 60 minutes as the discrete domains for IRMv1, REx, GroupDRO, and IIBNet. The data taken between 10:00 and 15:00 are as the testing set because the workload variance in this time period is the largest and it exhibits obviously unstable behavior. All the remaining data serves as the training set. Table 3 reports the performance of all methods. ERM performs the best in training but worst in testing, implying that ERM suffers from the distributional shift. On the other hand, IRMv1 performs the best across all existing methods, exceeding ERM by 9%. CIL outperforms ERM and IRMv1 by 13% and 5%, respectively, indicating its capability to recover a more invariant model achieving better OOD generalization.

### 4.2.4 WILDTIME-YEARBOOK

We adopt the Yearbook dataset in Wildtime benchmark (Yao et al., 2022)[*], which is a gender classification task on images taken from American high school students as shown in Figure 3. The Yearbook consists of 37K images with the time index as domains. The training set consists of data collected from 1930-1970 and the testing set covers 1970-2013. We adopt the same dataset processing, model architecture, and other settings with the WildTime. To keep consistent with Yao et al. (2022), we adopt the same baseline methods with Yao et al. (2022) and directly copy the performance of baseline methods from Yao et al. (2022).For the details of these baselines, we refer the reader to the Appendix B of Yao et al. (2022). Table 4 shows that we achieve the best OOD performance of 71.22%, improving about 1.5% over the previous SOTA methods (marked with underline).

## 5 CONCLUSION AND DISCUSSION

We proposed Continuous Invariance Learning (CIL) that extends invariance learning from discrete categorical indexed domains to natural continuous domain in this paper and theoretically demonstrated that CIL is able to learn invariant features on continuous domains under suitable conditions. However, learning invariance would be more challenging with larger DNNs due to IRM's inherent sensitivity to over-fitting (Lin et al., 2022a; Zhou et al., 2022). Recent works has shown the effectiveness of so called spurious feature diversification (Lin et al., 2023a), which has shown very promising performance even on modern large language models (Lin et al., 2023b). It would be an interesting future direction to explore feature diversification on continuous domains.

---

[*]https://github.com/huaxiuyao/Wild-Time

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

## A    LIMITATION AND SOCIAL IMPACT

**Limitation** Our framework uses min-max strategy which acquires the suboptimal solution in specific scenarios, and takes enough discrete time points as continuous domains in experiments without true continuous environments with infinity domain indices.

**Social Impact** We extend invariant learning into the continuous domains which is more natural to real-world tasks. Specifically, it would be helpful to solve the OOD problems related to time domains, e.g. it has been applied on Alipay Cloud to achieve the auto-scaling of server resources.

## B    RELATED WORK

**Causality, invariance, and distribution shift.** The invariance property is well known in causal literature back to early 1940s (Haavelmo, 1944), showing the conditional of the target given its direct causes are invariant under intervention on any node in the causal graph except for the target itself. In 2016, Invariant Causal Prediction (ICP) is first proposed in Peters et al. (2016) to utilize invariance property to identify the direct causes of the target. Models relying on the target's direct causes are robust to interventions. From the perspective of causality, the distributional shift in the testing distribution is due to the interventions on the causal graph (Arjovsky et al., 2019). So a model with invariance property can hopefully generalize even in the existence of distributional shifts. Anchor regression (Rothenhäusler et al., 2018) build the connection between distributional robustness with causality, and demonstrates that a suitable penalty in anchor regression based on the causal structure is equivalent to ensure a certain degree of robustness to distributional shift. Notably, Peters et al. (2016); Rothenhäusler et al. (2018) both assume the inputs $x$ are handcrafted meaningful features, which limits their applications in machine learning and deep learning where the input can be raw images.

Arjovsky et al. (2019) proposed the first invariance learning method, invariant risk minimization (IRM), which extends ICP (Peters et al., 2016) to deep learning by incorporating feature learning. Invariance learning has gained its popularity in recent years and inspired a line of excellent works, where a variety of variants have been proposed. To name a few, Krueger et al. (2021); Xie et al. (2020) penalize the variance of losses in domains; Ahuja et al. (2020) incorporates game theory into invariance learning; Chang et al. (2020) estimates the violation of invariance by training multiple independent networks; Jin et al. (2020) proposes an invariance penalty based on regret. Notably, these works all require discretely indexed domains. Another line of works try to learn invariant features when explicit domain indices are not provided (Creager et al., 2021). However, Lin et al. (2022b) theoretically shows that it is generally impossible to learn invariance without environmental information. Some recent works (Lin et al., 2022a; Zhou et al., 2022) show that invariance learning methods are sensitive to overfitting caused by the overparameterization of deep neural networks. These works are orthogonal to our study.

**Distribution Shift with Continuous Domains.** There are a few methods considering domain shifts with continuous domain indexes. Bobu et al. (2018); Hoffman et al. (2014); Wulfmeier et al. (2018); Bitarafan et al. (2016) consider the distribution shifts incrementally over time, while trying to perform domain adaptation sequentially. Continuously indexed domain adaption (CIDA) (Wang et al., 2020) is the first to adapt across (multiple) continuously indexed domains simultaneously. Note that these works all assume that *the input $x$ of the samples in the testing domains are available*, and are therefore *not* applicable for the OOD generalization tasks considered in this paper. Zhang et al. (2017b) tries to discover causal graph based on the continuous heterogeneity, but assumes that the features are given (cannot be learned from raw input), making it not applicable in our setting either.

## C    INVARIANCE PROPERTY IN CAUSALITY.

Figure 5 shows an example (similar to Figure 1 of ICP (Peters et al., 2016)) of an causal system with nodes $(y, x_1, x_2, x_3, x_4)$. Suppose our task is to build a model based on a subset of $\{x_1, x_2, x_3, x_4\}$ to predict $y$, under the distributional shifts in different domains caused by interventions. In our example, there are interventions on node $x_2$ in domain 2, and interventions on nodes $x_3$ and $x_4$ in domain 3. Notably, we do not allow interventions on target $y$ itself (indicating that the noise ratio of $y$ should be constant among domains). The invariance property shows that the conditional probability

of $y$ given its parents remains the same on interventions on any nodes except for the $y$ itself. In this example, $P(y|x_2, x_3)$ remains the same under all the three domains. However, one can check that $P(y|x_1)$ and $P(y|x_4)$ changes in domain 2 and 3, respectively. Therefore it is safe to build the model on $(x_2, x_3)$ to predict $y$. In contrast, $x_1$ and $x_4$ are unreliable because the conditional distribution is unstable under distributional shifts. In this example, $\boldsymbol{x}_v$ is $\{x_2, x_3\}$, and $\boldsymbol{x}_s$ is $\{x_1, x_4\}$.

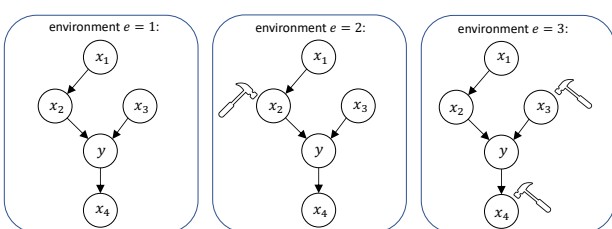

Figure 5: An illustration of the invariance property in causality (similar to Figure 1 in ICP (Peters et al., 2016)). This figure shows a causal system with five nodes, $y$, $x_1$, $x_2$, $x_3$, and $x_4$. Our task is to predict $y$ based on the $x'$s. There are different interventions in different domains, leading to distributional shifts. Intervention on a node can be simply interpreted as changing the node value. The changes can propagate to the descendants of the intervened node. The invariance property shows that $P(y|x_2, x_3)$ remains the same in all three domains. In contrast, $P(y|x_1)$ and $P(y|x_4)$ changes in domain 2 and 3 due to interventions, respectively. So it is safe to build model on $x_2$ and $x_3$ to predict $y$, which is expected to be stable under novel testing distribution.

## D DISCUSSIONS ON THE FIRST MOMENTS

To encourage conditional independence $\boldsymbol{y} \perp \boldsymbol{t}|\Phi(\boldsymbol{x})$, existing IRM variants try to align $\mathbb{E}^t[\boldsymbol{y}|\Phi(\boldsymbol{x})]$ (see Section 2.1) and our CIL proposes to align $\mathbb{E}^y[\boldsymbol{t}|\Phi(\boldsymbol{x})]$ (see Section 3).

### D.1 DISCRETE DOMAIN CASE

For discrete domain problems, when there are a sufficient number of samples in each domain $\boldsymbol{t}$ and class $\boldsymbol{y}$, $\mathbb{E}^y[\boldsymbol{t}|\Phi(\boldsymbol{x})]$ and $\mathbb{E}^t[\boldsymbol{y}|\Phi(\boldsymbol{x})]$ performs similarly (both reflecting the first moment of $\boldsymbol{y} \perp \boldsymbol{t}|\Phi(\boldsymbol{x})$, although from different perspectives), and neither of them has a clear advantage over the other. Consider the following example, which has gained popularity (Lin et al., 2022b): We have a binary classification task, $\boldsymbol{y} \in \{-1, 1\}$, containing 2 domains $\boldsymbol{t} = \{1, 2\}$, one invariant feature $\boldsymbol{x}_v$ and one spurious feature $\boldsymbol{x}_s$. Furthermore, we have

$$\boldsymbol{x}_v = \begin{cases} 1, \text{with prob } 0.5, \\ -1, \text{with prob } 0.5, \end{cases} \qquad \boldsymbol{y} = \begin{cases} \boldsymbol{x}_v, \text{with prob } 0.8, \\ -\boldsymbol{x}_v, \text{with prob } 0.2, \end{cases} \qquad \text{hold for both domains.} \qquad (3)$$

And

$$\text{In domain 1, } \boldsymbol{x}_s = \boldsymbol{y}, \text{in domain 2, } \boldsymbol{x}_s = \begin{cases} \boldsymbol{y}, \text{with prob } 0.5, \\ -\boldsymbol{y}, \text{with prob } 0.5, \end{cases} \qquad (4)$$

In other words, the correlation between the invariant feature $\boldsymbol{x}_v$ and the label $\boldsymbol{y}$ is always 0.8. The correlation between $\boldsymbol{x}_s$ and $\boldsymbol{y}$ is 1 in domain 1 and 0.5 in domain 2. By simple calculation, we have

$$\mathbb{E}^{t=1}[\boldsymbol{y}|\boldsymbol{x}_s = 1] = 1, \quad \mathbb{E}^{t=2}[\boldsymbol{y}|\boldsymbol{x}_s = 1] = 0$$
$$\mathbb{E}^{t=1}[\boldsymbol{y}|\boldsymbol{x}_s = -1] = -1, \quad \mathbb{E}^{t=2}[\boldsymbol{y}|\boldsymbol{x}_s = -1] = 0$$
$$\mathbb{E}^{t=1}[\boldsymbol{y}|\boldsymbol{x}_v = 1] = 1, \quad \mathbb{E}^{t=2}[\boldsymbol{y}|\boldsymbol{x}_v = 1] = 1$$
$$\mathbb{E}^{t=1}[\boldsymbol{y}|\boldsymbol{x}_v = -1] = -1, \quad \mathbb{E}^{t=2}[\boldsymbol{y}|\boldsymbol{x}_v = -1] = -1$$

and

$$\mathbb{E}^{y=1}[\boldsymbol{t}|\boldsymbol{x}_s = 1] = 4/3, \quad \mathbb{E}^{y=-1}[\boldsymbol{t}|\boldsymbol{x}_s = 1] = 2$$
$$\mathbb{E}^{y=1}[\boldsymbol{t}|\boldsymbol{x}_s = -1] = 2, \quad \mathbb{E}^{y=-1}[\boldsymbol{t}|\boldsymbol{x}_s = -1] = 4/3$$
$$\mathbb{E}^{y=1}[\boldsymbol{t}|\boldsymbol{x}_v = 1] = 3/2, \quad \mathbb{E}^{y=-1}[\boldsymbol{t}|\boldsymbol{x}_v = 1] = 3/2$$
$$\mathbb{E}^{y=1}[\boldsymbol{t}|\boldsymbol{x}_v = -1] = 3/2, \quad \mathbb{E}^{y=-1}[\boldsymbol{t}|\boldsymbol{x}_v = -1] = 3/2.$$

So we can observe that $\mathbb{E}^y[\boldsymbol{t}|\boldsymbol{x}_v]$ is the same for all $\boldsymbol{y}$, and $\mathbb{E}^t[\boldsymbol{y}|\boldsymbol{x}_v]$ is the same for all $\boldsymbol{t}$. Additionally, $\mathbb{E}^y[\boldsymbol{t}|\boldsymbol{x}_s]$ is different for $y = -1$ and $y = 1$, while $\mathbb{E}^t[\boldsymbol{y}|\boldsymbol{x}_s]$ is different for $t = 1$ and $t = 2$. In other words, if we aim to select a feature $\boldsymbol{x}$ from $\boldsymbol{x} \in \{\boldsymbol{x}_v, \boldsymbol{x}_s\}$ to align either $\mathbb{E}^y[\boldsymbol{t}|\boldsymbol{x}]$ or $\mathbb{E}^t[\boldsymbol{y}|\boldsymbol{x}_s]$, we would choose the invariant feature $\boldsymbol{x}_v$ in both cases. So our CIL method and existing methods performs similarly in this example with 2 domains.

### D.2 CONTINUOUS DOMAIN CASE

Consider a 10-class classification task with an infinite number of domains and each domain contains only one sample, i.e., $n \to \infty$, and $n/|\mathcal{T}| = 1$, Proposition 1 shows that REx would fail to identify the invariant feature with constant probability. Whereas, Theorem 2 shows that CIL can still effectively extract invariant features. Intuitively, existing methods aim to align $\mathbb{E}^t[\boldsymbol{y}|\Phi(\boldsymbol{x})]$ across different $\boldsymbol{t}$ values. However, in continuous environment settings with limited samples per domain, the empirical estimations $\hat{\mathbb{E}}^t[\boldsymbol{y}|\Phi(\boldsymbol{x})]$ become noisy. These estimations deviate significantly from the true $\mathbb{E}^t[\boldsymbol{y}|\Phi(\boldsymbol{x})]$, rendering existing methods ineffective in identifying invariant features. In contrast, CIL proposes to align $\mathbb{E}^y[\boldsymbol{t}|\Phi(\boldsymbol{x})]$, which can be accurately estimated as there are sufficient sample in each class.

## E ALGORITHM

---

**Algorithm 1** CIL: Continuous Invariance Learning

---

**Input:** Feature extractor $\Phi$, label classifier $\omega$, domain index regressor $h$ and $g$; The training dataset $\mathcal{D}^{\text{tr}} = \{(\mathbf{x}_i, \mathbf{y}_i, \mathbf{t}_i)_{i=1}^n\}$.
**Output:** The learned $\Phi$, classifier $\omega$, the domain regressor $h$ and $g$.
1: Initialize $\Phi$, $\omega$, $h$ and $g$.
2: **while** Not Converge **do**
3:     Sample a batch $\mathcal{B}$ from $\mathcal{D}^{\text{tr}}$.
4:     Obtain the loss for $g$ on the batch $\mathcal{B}$,
    $\mathcal{L}_{\mathcal{B}}(\Phi, g) = 1/|\mathcal{B}| \sum_{(\mathbf{x}, \mathbf{y}, \mathbf{t}) \in \mathcal{B}} \|g(\Phi(\mathbf{x}), \mathbf{y}), \mathbf{t}\|_2^2$.
5:     Perform one step of gradient ascent on $\mathcal{L}_{\mathcal{B}}(\Phi, g)$ w.r.t. $g$, i.e., $g \leftarrow g + \eta \nabla_g \mathcal{L}_{\mathcal{B}}(\Phi, g)$
6:     Obtain the loss for $(\Phi, \omega, h)$ on $\mathcal{B}$, $\mathcal{L}_{\mathcal{B}}(\omega, \Phi, h) = 1/|\mathcal{B}| \sum_{\mathbf{x}, \mathbf{y}, \mathbf{t} \in \mathcal{B}} (\ell(\omega(\Phi(\mathbf{x})), \mathbf{y}) + \lambda \|h(\Phi(\mathbf{x})), \mathbf{t}\|_2^2)$
7:     Perform one step of gradient descent on $\mathcal{L}_{\mathcal{B}}(\omega, \Phi, h)$ w.r.t. $(\omega, \Phi, h)$, i.e.,
    $(\omega, \Phi, h) \leftarrow (\omega, \Phi, h) + \eta \nabla_{(\omega, \Phi, h)} \mathcal{L}_{\mathcal{B}}(\omega, \Phi, h)$
8: **end while**
9: **return** $(\omega, \Phi, h, g)$

---

## F PROOF OF PROPOSITION 1

Recall that $\mathcal{R}^t(\boldsymbol{w}, \Phi) := \mathbb{E}^t[\ell(\boldsymbol{w}(\Phi(\boldsymbol{x})), \boldsymbol{y})]$ denotes the lose of $(\boldsymbol{w}, \Phi)$ in domain $t$, where $\boldsymbol{w}$ is the classifier $\Phi$ is the feature selector, and $\ell$ is the lose function. If there are many domains (i.e., $\mathcal{T}$ is large) and each domain only contains limited sample (i.e., $n/|\mathcal{T}|$ is limited), then the empirical REx

loss is as follows:

$$\hat{\mathcal{L}}(\Phi(\boldsymbol{x})) = \sum_{t \in \mathcal{T}} \hat{\mathcal{R}}^t(\omega, \Phi) + \lambda|\mathcal{T}|\mathrm{Var}(\hat{\mathcal{R}}^t(\omega, \Phi))$$

$$= \sum_{t \in \mathcal{T}} \mathcal{R}^t(\omega, \Phi) + \left( \sum_{t \in \mathcal{T}} \mathcal{R}^t(\omega, \Phi) - \sum_{t \in \mathcal{T}} \hat{\mathcal{R}}^t(\omega, \Phi) \right)$$

$$+ \lambda|\mathcal{T}| \left( -\left( \mathcal{R}^t(\omega, \Phi) - \hat{\mathcal{R}}^t(\omega, \Phi) \right) + \left( \mathcal{R}(\omega, \Phi) - \hat{\mathcal{R}}(\omega, \Phi) \right) + \left( \mathcal{R}^t(\omega, \Phi) - \mathcal{R}(\omega, \Phi) \right) \right)^2$$

$$= \sum_{t \in \mathcal{T}} \mathcal{R}^t(\omega, \Phi) + \lambda \underbrace{\sum_{t \in \mathcal{T}} \left( \mathcal{R}^t(\omega, \Phi) - \mathcal{R}(\omega, \Phi) \right)^2}_{A_0}$$

$$+ \underbrace{\sum_{t \in \mathcal{T}} \left( \mathcal{R}^t(\omega, \Phi) - \hat{\mathcal{R}}^t(\omega, \Phi) \right)}_{A_1} + \underbrace{\sum_{t \in \mathcal{T}} \left( \mathcal{R}^t(\omega, \Phi) - \hat{\mathcal{R}}^t(\omega, \Phi) \right)^2}_{A_2} + \underbrace{\sum_{t \in \mathcal{T}} \left( \mathcal{R}(\omega, \Phi) - \hat{\mathcal{R}}(\omega, \Phi) \right)^2}_{A_3}$$

$$- \underbrace{\sum_{t \in \mathcal{T}} 2 \left( \mathcal{R}^t(\omega, \Phi) - \hat{\mathcal{R}}^t(\omega, \Phi) \right) \left( \mathcal{R}(\omega, \Phi) - \hat{\mathcal{R}}(\omega, \Phi) \right)}_{A_4}$$

$$- \underbrace{\sum_{t \in \mathcal{T}} 2 \left( \mathcal{R}^t(\omega, \Phi) - \hat{\mathcal{R}}^t(\omega, \Phi) \right) \left( \mathcal{R}^t(\omega, \Phi) - \mathcal{R}(\omega, \Phi) \right)}_{A_5}$$

$$+ \underbrace{\sum_{t \in \mathcal{T}} 2 \left( \mathcal{R}(\omega, \Phi) - \hat{\mathcal{R}}(\omega, \Phi) \right) \left( \mathcal{R}^t(\omega, \Phi) - \mathcal{R}(\omega, \Phi) \right)}_{A_6}$$

We then have

$$A_0 = \begin{cases} \sigma_R \chi(|\mathcal{T}|), \text{ if } \Phi = \Phi_s \\ 0, \text{ if } \Phi = \Phi_v \\ 0, \text{ if } \Phi = \Phi_{null} \end{cases}$$

$$A_1 = \sum_{t \in \mathcal{T}} \sum_{i=1}^{n_e} \epsilon_i \sim \sigma/\sqrt{n}\mathcal{N}(0, 1)$$

$$A_2 = \sum_t (\sum_{i=1}^{n_e} \epsilon_i)^2 \sim \sigma/\sqrt{n_e}\chi(|\mathcal{T}|)$$

$$A_3 = \sum_t (\sum_{i=1}^{n} \epsilon_i)^2 \sim \sigma/\sqrt{n}\chi(|\mathcal{T}|)$$

$$A_4 = 2 \left( \sum_{t \in \mathcal{T}} \left( \mathcal{R}^t(\omega, \Phi) - \hat{\mathcal{R}}^t(\omega, \Phi) \right) \right) \left( \mathcal{R}(\omega, \Phi) - \hat{\mathcal{R}}(\omega, \Phi) \right)$$

$$= 2\frac{1}{n} \sum_{i=1}^{n} \epsilon_i \frac{1}{n} \sum_{i=1}^{n} \epsilon_i \sim 2\frac{\sigma}{\sqrt{n}}\chi(1)$$

$$A_5 \sim \sqrt{\sigma^2/\sqrt{n_e} + \sigma_R^2}\chi(|\mathcal{T}|) - \sqrt{\sigma^2/\sqrt{n_e} + \sigma_R^2}\chi(|\mathcal{T}|)$$

$$A_6 = 2 \left( \mathcal{R}(\omega, \Phi) - \hat{\mathcal{R}}(\omega, \Phi) \right) \sum_{t \in \mathcal{T}} \left( \mathcal{R}^t(\omega, \Phi) - \mathcal{R}(\omega, \Phi) \right) = 2 \left( \mathcal{R}(\omega, \Phi) - \hat{\mathcal{R}}(\omega, \Phi) \right) \times 0 = 0$$

By taking account $n_e = n/|\mathcal{T}|$, we have

$$\mathbb{E}[\hat{\mathcal{L}}(\Phi(\boldsymbol{x}))|\sigma_s, \sigma_v, \sigma_{null}, \sigma_R] = \begin{cases} \sum_{t \in \mathcal{T}} \mathcal{R}^t(\omega, \Phi_s) + \sigma_R|\mathcal{T}| + \frac{\sigma_s}{\sqrt{n}}(2 + |\mathcal{T}| + |\mathcal{T}|^2) \\ \sum_{t \in \mathcal{T}} \mathcal{R}^t(\omega, \Phi_v) + \frac{\sigma_v}{\sqrt{n}}(2 + |\mathcal{T}| + |\mathcal{T}|^2) \\ \sum_{t \in \mathcal{T}} \mathcal{R}^t(\omega, \Phi_{null}) + \frac{\sigma_{null}}{\sqrt{n}}(2 + |\mathcal{T}| + |\mathcal{T}|^2) \end{cases}$$

Denote $Q = \left(\sum_{t\in\mathcal{T}}\mathcal{R}^t(w,\Phi_v) - \sum_{t\in\mathcal{T}}\mathcal{R}^t(w,\Phi_s)\right)$, assume $\delta_R|\mathcal{T}| \geq Q$. we have $\mathbb{E}[\hat{\mathcal{L}}(\Phi_v(\boldsymbol{x}))|\sigma_s,\sigma_v,\sigma_R] > \mathbb{E}[\hat{\mathcal{L}}(\Phi_s(\boldsymbol{x}))|\sigma_s,\sigma_v,\sigma_R]$ if

$$(\sigma_v - \sigma_s) \geq \frac{2\sigma_R\sqrt{n}}{|\mathcal{T}|} \geq \frac{\sqrt{n}(\sigma_R|\mathcal{T}|+|Q|)}{|\mathcal{T}|^2} \geq \frac{\sqrt{n}(\sigma_R|\mathcal{T}|+|Q|)}{(2+|\mathcal{T}|+|\mathcal{T}|^2)}.$$

Since $\delta_s, \delta_v$ are independently drawn from a hyper exponential distribution where the density function $P(x;\lambda) = \lambda\exp(-\lambda x)$, so $P(\delta_v - \delta_s \leq z) = 1 - 1/2\exp(-\lambda z)$. Then if

$$|\mathcal{T}| \geq \frac{\sigma_R\sqrt{n}}{\Delta\mathcal{G}^{-1}(1/4)},$$

REx is unable to identify the invariant feature with a probability of at least 1/4.

## G  PROOF FOR LEMMA 1

*Proof.*

$$\arg\min_{h\in\mathcal{H}}\mathbb{E}_{(\boldsymbol{x},t)}(h(\Phi(\boldsymbol{x}))-\boldsymbol{t})^2$$
$$=\arg\min_{h\in\mathcal{H}}\mathbb{E}_{(\Phi(\boldsymbol{x}),t)}(h(\Phi(\boldsymbol{x}))-\boldsymbol{t})^2$$
$$=\arg\min_{h\in\mathcal{H}}\mathbb{E}_{\Phi(\boldsymbol{x})}\mathbb{E}_{\boldsymbol{t}\sim\mathbb{P}(\boldsymbol{t}|\Phi(\boldsymbol{x}))}(h(\Phi(\boldsymbol{x}))-\boldsymbol{t})^2.$$

Because

$$\mathbb{E}_{\boldsymbol{t}\sim\mathbb{P}(\boldsymbol{t}|\Phi(\boldsymbol{x}))}(h(\Phi(\boldsymbol{x}))-\boldsymbol{t})^2$$
$$=h(\Phi(\boldsymbol{x}))^2 - 2h(\Phi(\boldsymbol{x}))^\top\mathbb{E}[\boldsymbol{t}|\Phi(\boldsymbol{x})] + \mathbb{E}[\boldsymbol{t}^2|\Phi(\boldsymbol{x})],$$

we then know the minimum is achieved at $h(\Phi(\boldsymbol{x})) = \mathbb{E}[\boldsymbol{t}|\Phi(\boldsymbol{x})] = h^*(\Phi(\boldsymbol{x}))$ by solving this quadratic problem. The minimum loss achieved by $h^*(\cdot)$ is

$$\mathbb{E}_{\Phi(\boldsymbol{x})}\left[\mathbb{E}[\boldsymbol{t}^2|\Phi(\boldsymbol{x})] - (\mathbb{E}[\boldsymbol{t}|\Phi(\boldsymbol{x})])^2\right] = \mathbb{E}_{\Phi(\boldsymbol{x})}[\mathbb{V}[\boldsymbol{t}|\Phi(\boldsymbol{x})]],$$

We can prove the result for $g^*(\cdot)$ in a similar way, and the minimum loss achieved by $g^*(\cdot)$ is $\mathbb{E}_{\Phi(\boldsymbol{x}),\boldsymbol{y}}[\mathbb{V}[\boldsymbol{t}|\Phi(\boldsymbol{x}),\boldsymbol{y}]]$. $\square$

## H  PROOF FOR THEOREM 2

*Proof.* The penalty term of Eqn equation 2 is

$$\min_{h\in\mathcal{H}}\max_{g\in\mathcal{G}}\mathbb{E}_{\boldsymbol{x},\boldsymbol{y},\boldsymbol{t}}\left[(h(\Phi(\boldsymbol{x}))-\boldsymbol{t})_2^2 - (g(\Phi(\boldsymbol{x}),\boldsymbol{y})-\boldsymbol{t})_2^2\right]$$
$$=\mathbb{E}_{\Phi(\boldsymbol{x})}[\mathbb{V}[\boldsymbol{t}|\Phi(\boldsymbol{x})]] - \mathbb{E}_{\Phi(\boldsymbol{x}),\boldsymbol{y}}[\mathbb{V}[\boldsymbol{t}|\Phi(\boldsymbol{x}),\boldsymbol{y}]]$$
$$=\mathbb{E}_{\Phi(\boldsymbol{x}),\boldsymbol{y}}[\mathbb{E}[\boldsymbol{t}|\Phi(\boldsymbol{x}),\boldsymbol{y}]^2] - \mathbb{E}_{\Phi(\boldsymbol{x})}[\mathbb{E}[\boldsymbol{t}|\Phi(\boldsymbol{x})]^2]$$
$$=\mathbb{E}_{\Phi(\boldsymbol{x}),\boldsymbol{y}}[\mathbb{E}[\boldsymbol{t}|\Phi(\boldsymbol{x}),\boldsymbol{y}]^2] - \mathbb{E}_{\Phi(\boldsymbol{x})}[(\mathbb{E}_{\boldsymbol{y}}\mathbb{E}[\boldsymbol{t}|\Phi(\boldsymbol{x}),\boldsymbol{y}])^2]$$
$$\geq 0.$$

The last inequality is due to Jensen's inequality and the convexity of the quadratic function. The inequality is achieved only when $\mathbb{E}[\boldsymbol{t}|\Phi(\boldsymbol{x})] = \mathbb{E}[\boldsymbol{t}|\Phi(\boldsymbol{x}),\boldsymbol{y}], \forall\boldsymbol{y}\in\mathcal{Y}$. $\square$

## I  PROOF FOR PROPOSITION 2

Denote $\theta = [w,\Phi,h]$ and $\hat{\theta} = [\hat{w},\hat{\Phi},\hat{h}]$. We further use $q(\theta,g;\mathbf{z})$ denote the loss of Eqn 2 on a single sample $\mathbf{z} := [\mathbf{x},\mathbf{y},\mathbf{z}]$. We start by stating some common assumptions in theoretical analysis as follows: Our goal is to analyze the performance of the approximate solution $(\hat{w},\hat{\Phi},\hat{h},\hat{g})$ on the training dataset with finite samples.

**Assumption 4** ((Li & Liu, 2021)). *Let $\bar{\mathcal{D}}^{tr}$ be the dataset generated by replacing one data point in the training dataset $\mathcal{D}^{tr}$ with another data point drawn independently from the training distribution. We assume SGDA is $\epsilon$-argument-stable if for any $\mathcal{D}^{tr}$ such that*

$$\|\theta_{SGDA}(\bar{\mathcal{D}}^{tr}) - \theta_{SGDA}(\mathcal{D}^{tr})\| \leq \epsilon, \quad \|g_{SGDA}(\bar{\mathcal{D}}^{tr}) - g_{SGDA}(\mathcal{D}^{tr})\| \leq \epsilon \tag{5}$$

**Assumption 5.** *[(Li & Liu, 2021)] Denoting $(w, \Phi, h)$ as $\theta$ and $Q(w, \Phi, h, g)$ as $Q(\theta, g)$ for short, $Q(\theta, g)$ is $\mu$-strongly convex in $\theta$, i.e.,*

$$Q(\theta_1, g) - Q(\theta_2, g) \geq \nabla_\theta Q(\theta_2, g)^\top (\theta_1 - \theta_2) + \frac{\mu}{2}\|\theta_1 - \theta_2\|_2^2,$$

*and $\mu$-strongly concave in $g$, i.e.,*

$$Q(\theta_1, g) - Q(\theta_2, g) \leq \nabla_\theta Q(\theta_2, g)^\top (\theta_1 - \theta_2) - \frac{\mu}{2}\|\theta_1 - \theta_2\|_2^2.$$

The convex-concave assumption for the minimax problem is popular in existing literature (Li & Liu, 2021; Lei et al., 2021; Farnia & Ozdaglar, 2021; Zhang et al., 2021), simply because non-convex-non-concave minimax problems are extremely hard to analyze due to their non-unique saddle points. When $w$ and $h$ (the classifier for $\boldsymbol{y}$ and regressor for $\boldsymbol{t}$) are linear, it is easy to verify that $Q$ is convex in them. Furthermore, recent theoretical studies show that the overparameterized neural networks (NN) behave like convex systems and training large NN is likely to converge to the global optimum (Jacot et al., 2018; Mei et al., 2018).

**Assumption 6** (Lipschitz continuity (Li & Liu, 2021)). *Let $L > 0$. Assume that for any $\theta$, $g$ and $\mathbf{z}$, we have*

$$\|\nabla_\theta q(\theta, g; \mathbf{z})\| \leq L \quad and \quad \|\nabla_g q(\theta, g; \mathbf{z})\| < L.$$

**Assumption 7** (Smoothness (Li & Liu, 2021)). *Let $\beta > 0$. Assume that for any $\theta_1, \theta_2, g_1, g_2$ and $\mathbf{z}$, we have*

$$\left\|\begin{pmatrix} \nabla_\theta f(\theta_1, g_1; \mathbf{z}) - \nabla_\theta f(\theta_2, g_2; \mathbf{z}) \\ \nabla_g f(\theta_1, g_1; \mathbf{z}) - \nabla_g f(\theta_2, g_2; \mathbf{z}) \end{pmatrix}\right\| \leq \beta \left\|\begin{pmatrix} \theta_1 - \theta_2 \\ g_1 - g_2 \end{pmatrix}\right\|.$$

We define the strong primal-dual empirical risk as follows:

$$\Delta_{SGDA}^s(\hat{\theta}, \hat{g}) = \sup_g \hat{Q}(\hat{\theta}, g) - \inf_\theta \hat{Q}(\theta, \hat{g})$$

We first restate Theorem 2 as follows:

**Theorem 3.** *Assume we solve equation 2 by SGDA as introduced in Algorithm 1 and obtain $(\epsilon_1, \epsilon_2)$ solution $(\hat{\theta}, \hat{h})$. Further, suppose Assumption 5 holds and SGDA is $\epsilon$-argument-stable as described in Assumption 4, then for any $\delta > 0$, fix $\eta > 0$, we have with probability at least $1 - \delta$*

$$Q^*(\hat{\theta}) \leq (1 + \eta)\inf_\theta Q^*(\theta) + C\frac{1 + \eta}{\eta}\left(\frac{M}{n}\log\frac{1}{\delta} + \left(\frac{\beta}{\mu} + 1\right)L\epsilon\log_2 n\log\frac{1}{\delta} + \epsilon_1 + \epsilon_2\right)$$

*where $\mu$ is the strongly-convexity, $L$ is the Lipschitz, $\epsilon$ is the stability, $M$ is $n$ is the sample size of the training dataset, $\tilde{O}$ absorbs logarithmic and constant variables which is specified in the appendix.*

*Proof.* By the suboptimality assumption of $\hat{\theta}$ and $\hat{g}$, we have

$$\begin{aligned}
\Delta_{SGDA}^s(\hat{\theta}, \hat{g}) &= \sup_g \hat{Q}(\hat{\theta}, g) - \inf_\theta \hat{Q}(\theta, \hat{g}) \\
&\leq \hat{Q}(\hat{\theta}, \hat{g}) + \epsilon_2 - (\hat{Q}(\hat{\theta}, \hat{g}) - \epsilon_1) \\
&= \epsilon_1 + \epsilon_2
\end{aligned}$$

Denote $\theta^* = \arg\min_\theta Q^*(\theta)$ and $\hat{g}^* = \arg\max_g Q(\hat{\theta}, g)$. We can decompose

$$Q^*(\hat{\theta}) - \inf_\theta Q^*(\theta) = \underbrace{Q^*(\hat{\theta}) - \hat{Q}^*(\hat{\theta})}_{A_1} + \underbrace{\hat{Q}^*(\hat{\theta}) - \hat{Q}(\theta^*, \hat{g})}_{A_2} + \underbrace{\hat{Q}(\theta^*, \hat{g}) - Q(\theta^*, \hat{g})}_{A_3} + \underbrace{Q(\theta^*, \hat{g}) - Q^*(\theta^*)}_{A_4}$$

We bound $A_1 - A_4$ respectively:

- From Eqn (22) of Li & Liu (2021), we have

$$A_1 \leq \frac{2M \log(3/\delta)}{3n} + 50\sqrt{2}\epsilon e L \frac{\beta + \mu}{\mu} \log_2 nlog(3e/\delta) + \tag{6}$$

$$\sqrt{\frac{\left(4MQ(\hat{\theta}, \hat{g}^*) + 1/2(\beta/\mu + 1)^2 L^2 \epsilon^2 + 32n(\beta/\mu + 1)^2 L^2 \epsilon^2 \log(3/\delta)\right) \log(3/\delta)}{n}} \tag{7}$$

- we have

$$A_2 = \sup \hat{Q}(\hat{\theta}, g) - \hat{Q}(\theta^*, \hat{g}) \tag{8}$$

$$\leq \sup \hat{Q}(\hat{\theta}, g) - \inf_{\theta} \hat{Q}(\theta, \hat{g}) \tag{9}$$

$$\leq \epsilon_1 + \epsilon_2 \tag{10}$$

- By Eqn (10) of Li & Liu (2021),

$$A_3 = \hat{Q}(\theta^*, \hat{g}) - Q(\theta^*, \hat{g}) \leq \frac{2M \log(3/\delta)}{3n} + 50\sqrt{2}e\epsilon \log_2 n \log(3e/\delta) \tag{11}$$

$$+ \sqrt{\frac{(4MQ(\theta^*, \hat{h}) + \epsilon^2/2 + 32n\epsilon^2 \log(3/\delta)) \log(3/\delta)}{n}} \tag{12}$$

- At last

$$Q(\theta^*, \hat{g}) - Q^*(\theta^*) = Q(\theta^*, \hat{g}) - \sup_g Q(\theta^*, g) \leq 0. \tag{13}$$

Putting these together with some rearrangement, we finally have

$$Q^*(\hat{\theta}) \leq (1 + \eta) \inf_{\theta} Q^*(\theta) + C \frac{1 + \eta}{\eta} \left( \frac{M}{n} \log \frac{1}{\delta} + \left( \frac{\beta}{\mu} + 1 \right) L\epsilon \log_2 n \log \frac{1}{\delta} + \epsilon_1 + \epsilon_2 \right),$$

where $C$ is a constant. the description of the CIL method in the introduction section is hard to understand. □

## J  DETAIL DESCRIPTION ABOUT SETTING OF EMPIRICAL VERIFICATION

In this section, we show the spurious relationship $p_s(t)$ varies across the domains as shown in Figure 6.

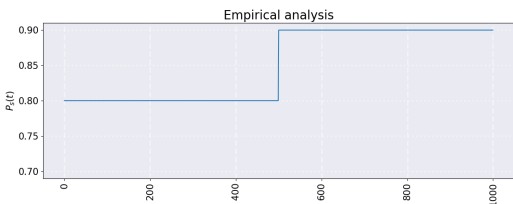

Figure 6: Spurious Relation of Empirical Verification 3.1

## K  LOGIT DATASET EXPERIMENT

**Setting.** We generate the first synthetic dataset with the invariant feature $x_v \in \mathbb{R}^2$, spurious feature $x_s \in \mathbb{R}^{20}$ and target $y \in \{0, 1\}$. The continuous domains $t \in [0, 100]$. The conditional distribution of $x_s$ given $y$ varies along $t$ as follows:

$$y = \begin{cases} 0, & \text{w.p. } 0.5, \\ 1, & \text{w.p. } 0.5, \end{cases} \quad x_v \sim \begin{cases} \mathcal{N}(y, \sigma^2), & \text{w.p. } p_v, \\ \mathcal{N}(-y, \sigma^2), & \text{w.p. } 1 - p_v, \end{cases} \quad x_s \sim \begin{cases} \mathcal{N}(y, \sigma^2), & \text{w.p. } p_s(t), \\ \mathcal{N}(-y, \sigma^2), & \text{w.p. } 1 - p_s(t), \end{cases}$$

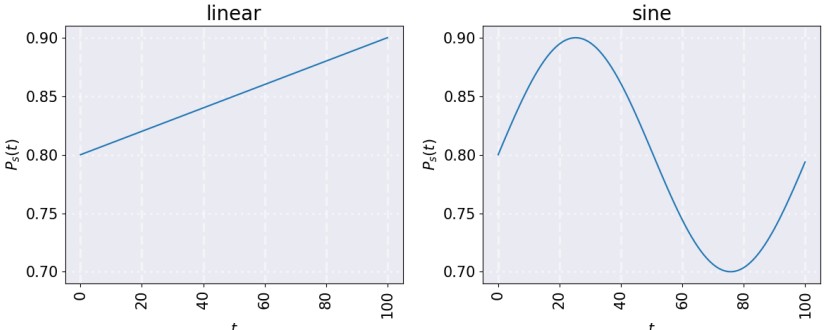

Figure 7: Spurious correlation of logit dataset

| Env. Type | Split Num | Method | Linear $p_s$ | Sine $p_s$ |
|---|---|---|---|---|
| None | – | ERM | 25.33(2.01) | 36.18(3.06) |
| Discrete | 4 | IRMv1 | 55.39(6.87) | 69.91(2.40) |
| | 8 | REx | 46.46(7.50) | 68.15(5.51) |
| | 2 | GroupDRO | 46.68(1.80) | 60.95(2.23) |
| | 16 | IIBNet | 51.50(18.60) | 49.68(21.87) |
| Continuous | – | IRMv1 | 25.90(2.85) | 41.28(3.66) |
| | – | REx | 42.73(11.47) | 63.33(5.05) |
| | – | Diversify | 53.39(2.96) | 53.30(1.93) |
| | – | CIL | **60.95(6.59)** | **76.25(4.99)** |

Table 4: Comparison on the synthetic Logit data. The metrics including accuracy and standard deviations in parenthesis are calculated on three independent runs. The environment type "Discrete" indicates that we manually create by equally splitting the raw continuous domains. The environment type "Continuous" indicates we use the original continuous domain index. "Split Num" stands for the number of domains we manually create and we report the best performance among spilt $\{2, 4, 8, 16\}$.

where $p_v$ and $p_s(t)$ are the probabilities of the feature's agreement with label $y$.

Notably, $p_s(t)$ varies among domains while $p_v$ stays invariant. The observed feature $x$ is a concatenation of $x_v$ and $x_s$, i.e., $x := [x_v, x_s]$. We generate 2000 samples as the training dataset $\{(\mathbf{x}_i, \mathbf{y}_i, \mathbf{t}_i)\}_{i=1}^{2000}$. Our goal is to learn a model $f(x)$ to predict $y$ that solely relies on $x_v$. We set $p_v$ to be 0.9 in all domains. $p_s(t)$ varies across the domains as in Figure 7 .

We can see that $x_s$ exhibits a high but unstable correlation with $y$. Since existing methods need discrete domains, we equally divide the continuous domain into different numbers of discrete domains, i.e, $\{2, 4, 8, 16\}$.

**Results.** Table 4 shows the test accuracy of each method. ERM performs the worst and the large gap implies that ERM models heavily depend on spurious features. IRMv1 improves by 25% on average compared to ERM. CIL outperforms ERM by over 30% on average. CIL also improves by 5-7% over all existing methods based on discrete domains on the best manual discrete domain partition, which indicates that CIL can learn invariant features more effectively. The trend of how the performance of IRMv1 and REx changes with the split number is shown in Table 5

## L ADDITIONAL EXPERIMENT RESULTS FOR 4.1.1

**Details on continous CMNIST**. The original CMNIST dataset consists of two domains with varying spurious correlation values: 0.9 in one domain and 0.8 in the other. To simulate a continuous problem, we randomly assign time indices 1-512 to samples in the first domain and indices 513-1024 to the second domain. The spurious correlation only changes at time index 513, as shown in Figure 5 on Page 16 of the Appendix file. In this case, each of the 1024 domains comprises approximately 50

| Env. Type | Split Num | Method | Linear $p_s$ | Sine $p_s$ |
|---|---|---|---|---|
| None | – | ERM | 25.33(2.01) | 36.18(3.06) |
| Discrete | 2 | IRMv1 | 53.11(1.92) | 66.16(4.29) |
| | 4 | | 55.39(6.87) | 69.91(2.40) |
| | 5 | | 39.35(11.70) | 55.98(3.91) |
| | 8 | | 47.90(2.44) | 59.03(5.32) |
| | 16 | | 30.66(4.45) | 56.90(4.98) |
| | 50 | | 27.32(0.97) | 42.73(11.47) |
| | 100 | | 25.90(2.85) | 41.28(13.66) |
| | 2 | REx | 39.69(4.56) | 59.76(1.22) |
| | 4 | | 55.80(12.49) | 64.05(11.29) |
| | 5 | | 50.77(11.69) | 65.15(5.77) |
| | 8 | | 46.46(0.75) | 68.15(5.51) |
| | 16 | | 44.05(9.53) | 67.78(6.60) |
| | 50 | | 42.73(11.47) | 61.18(2.78) |
| | 100 | | 41.87(0.48) | 63.33(5.05) |
| Continuous | – | CIL | **60.95(6.59)** | **76.25(4.99)** |

Table 5: Comparison on the synthetic Logit data. The metric is accuracy and standard deviations in parenthesis are calculated on three independent runs. Split Num stands for the number of domains we manually create by equally splitting the raw continuous domains. We try settings where the spurious correlation $p_s(t)$ changes by linear and Sine functions, respectively.

| Env. Type | Split Num | Method | Train | Test |
|---|---|---|---|---|
| Discrete | 2 | IRMv1 | 49.74(0.62) | 50.01(0.34) |
| | 4 | | 51.02(0.86) | **49.72(0.86)** |
| | 8 | | **51.45(3.53)** | 49.46(0.94) |
| | 16 | | 51.31(2.28) | 49.71(0.90) |
| | 100 | | 50.21(1.96) | 49.35(0.05) |
| | 500 | | 49.56(0.74) | 47.02(2.72) |
| | 1000 | | 49.67(0.33) | 48.70(2.65) |
| | 2 | REx | 83.09(0.40) | 45.50(1.12) |
| | 4 | | **82.75(0.33)** | 47.14(2.22) |
| | 8 | | 82.05(0.67) | **49.31(2.55)** |
| | 16 | | 82.32(0.45) | 47.86(1.54) |
| | 100 | | 80.54(0.78) | 49.12(0.20) |
| | 500 | | 79.21(0.45) | 42.55(1.05) |
| | 1000 | | 78.98(0.32) | 41.87(0.48) |

Table 6: Accuracy on the continuous CMNIST with $p_s(t)$ as a linear function with different split number in (2, 4, 8, 16, 100, 500, 1000).

samples. Based on the results in Figure 2 in the main part of the manuscript, REx and IRMv1 display testing accuracies close to random guessing in this scenario. Similar constructions are made for 4, 8, ..., 512 domains.

In this section, we provide the complete experiment results in Table 6 and Table 7 for IRMv1, REx, GroupDRO, and IIBNet with other numbers of splits ("split num") on the continuous CMNIST dataset. The settings are the same as in Section 4.1.1, where Table 2 show the best results for each method with one corresponding "split num".

**Results**. Table 6 and 7 report the training and testing accuracy of different domain splitting schemes (2, 4, 8, 16) for both IRMv1 and REx on the continuous CMNIST dataset with two $p_s(t)$ settings. Under linear $p_s(t)$, REx performs the best with split number 8 by improving around 4% over the worst one. However, the results are fairly close under sine $p_s(t)$. On the other hand, IRMv1 with all testing accuracy around 0.5 does not seem to be able to extract useful features for the task with either linear $p_s(t)$ or sine $p_s(t)$.

## M  ADDITIONAL RESULTS FOR THE EXPERIMENTS 4.2

The result across different split number is shown in Table 8

| Env. Type | Split Num | Method | Train | Test |
|---|---|---|---|---|
| Discrete | 2 | IRMv1 | 49.47(0.21) | 49.44(0.57) |
| | 4 | | **49.77(0.78)** | 49.96(0.55) |
| | 8 | | 49.60(0.65) | 49.85(0.26) |
| | 16 | | 49.74(0.62) | **50.01(0.34)** |
| | 100 | | 49.62(0.58) | 49.83(0.63) |
| | 500 | | 50.28(0.87) | 46.23(10.11) |
| | 1000 | | 50.43(1.23) | 49.63(15.06) |
| | 2 | REx | **82.95(0.47)** | 54.17(1.37) |
| | 4 | | 81.93(0.91) | **54.97(1.71)** |
| | 8 | | 81.59(0.89) | 54.96(2.10) |
| | 16 | | 81.60(0.69) | 54.07(1.79) |
| | 100 | | 80.94(0.78) | 48.66(0.79) |
| | 500 | | 81.48(0.67) | 43.14(0.97) |
| | 1000 | | 79.97(0.79) | 42.24(0.74) |

Table 7: Accuracy on the continuous CMNIST with $p_s(t)$ as a Sine function with different split number in (2, 4, 8, 16, 100, 500, 1000).

| Env. Type | Split Num | Method | HousePrice | Insurance Fraud |
|---|---|---|---|---|
| Discrete | 2 | IRMv1 | 72.68(0.36) | 71.52(1.35) |
| | 5 | | 73.46(1.41) | 68.41(1.13) |
| | 10 | | 72.97(1.88) | 67.28(1.64) |
| | 25 | | 74.42(2.10) | 54.25(1.75) |
| | 50 | | 75.40(0.99) | 52.09(2.05) |
| | 2 | REx | 70.97(0.06) | 74.43(1.32) |
| | 5 | | 71.30(1.17) | 74.42(0.87) |
| | 10 | | 70.46(0.25) | 73.20(1.65) |
| | 25 | | 71.01(1.30) | 72.96(0.24) |
| | 50 | | 68.82(0.92) | 72.90(0.46) |
| Continuous | – | CIL | **60.95(6.59)** | **76.25(4.99)** |

Table 8: Comparison on the real world dataset: HousePrice and Insurance. The metric is accuracy and standard deviations in parenthesis are calculated on three independent runs. Split Num stands for the number of domains we manually create by equally splitting the raw continuous domains.

# N    ABLATION STUDY

In this section, we tried different penalty set up in equation 2 on the auto-scaling dataset to validate the robustness of CIL. The approximating functions $h(\Phi(x)), g(\Phi(x), y)$ are implemented by 2-Layer MLPs. We evaluate the performance changes by increasing either the hidden dimension of the MLPs (Table 9) or penalty weight $\lambda$ (Table 10).

| Hidden Dimension | ID Accuracy | OOD Accuracy |
|---|---|---|
| 32 | 84.37(5.95) | 64.24(5.47) |
| 64 | 84.17(2.58) | 70.40(1.18) |
| 128 | 81.25(1.65) | **71.29(0.04)** |
| 256 | **85.57(2.67)** | 68.12(2.25) |
| 512 | 85.28(2.27) | 68.77(1.60) |

Table 9: ID and OOD accuracy on the auto-scaling dataset across different MLP setups for $h(\Phi(x)), g(\Phi(x), y)$ in equation 2. Standard deviation in brackets is calculated with 3 independent runs. Other settings are kept the same as in Section O

**Results**.    The ID and OOD accuracy of different MLP setup (hidden dimension) for $h(\Phi(x)), g(\Phi(x), y)$ under CIL on the auto-scaling dataset are shown in Table 9. CIL performs the best when the hidden dimension is 64, and the performance is stable with even higher dimensions. However, the testing accuracy drops to 64% when the dimension is reduced to 32, as the model is probably too simple to be able to extract enough invariant features. But it is still better than ERM shown in Table 3.

| Penalty Weight | Train | Test |
|---|---|---|
| 100 | **86.91(1.31)** | 67.60(1.26) |
| 1000 | 84.47(2.57) | 70.01(1.54) |
| 10000 | 84.17(2.58) | 70.40(1.18) |
| 100000 | 83.95(2.73) | **70.42(1.16)** |
| 1000000 | 83.95(2.73) | 70.35(1.24) |

Table 10: ID and OOD accuracy on the auto-scaling dataset across different penalty weights $\lambda$ in equation 2. Standard deviation in brackets is calculated with 3 independent runs. Other settings are kept the same as in Section O

| Dataset | LR | OLR | Steps | Penalty Step | Penalty Weight |
|---|---|---|---|---|---|
| Logit (linear) | 0.001 | 0.001 | 1500 | 500 | 10000 |
| Logit (sine) | 0.001 | 0.001 | 1500 | 500 | 10000 |
| CMNIST (linear) | 0.001 | 0.001 | 1000 | 500 | 8000 |
| CMNIST (sine) | 0.001 | 0.001 | 1000 | 500 | 8000 |
| HousePrice | 0.001 | 0.01 | 1000 | 500 | 100000 |
| Insurance | 0.001 | 0.01 | 1500 | 500 | 10000 |
| Auto-scaling | 0.001 | 0.01 | 1000 | 500 | 10000 |
| WildTime-YearBook | 0.00001 | 0.001 | 1000 | 500 | 100 |

Table 11: The running setup for our CIL on each Dataset

Similarly, Table 10 reports the training and testing accuracy for different penalty weights $\lambda$. They all outperform ERM and the discrete methods in Table 9. All accuracy is close to 70% when the penalty weight is larger than or equal to 1000, which shows the model is robust to different penalty weights. The performance improvement lowers to 67% when the penalty is reduced to 100, but it still outperforms ERM by 10%. The above experiments prove the robustness of our CIL, which can improve the model performance in any setup case.

## O    SETTINGS OF EXPERIMENTS

In this section, we provide the training and hyperparameter details for the experiments. All experiments are done on a server base on Alibaba Group Enterprise Linux Server release 7.2 (Paladin) system which has 2 GP100GL [Tesla P100 PCIe 16GB] GPU devices.

- **LR**: learning rate of the classification model $\Phi(\boldsymbol{x})$), e.g. 1e-3.
- **OLR**: learning rate of the penalty model $h(\Phi(\boldsymbol{x})), g(\Phi(\boldsymbol{x}), \boldsymbol{y})$, e.g. 0.001
- **Steps**: total number of epochs for the training process, e.g. 1500
- **Penalty Step**: number of epochs when to introduce penalty, e.g. 500
- **Penalty Weight**: the invariance penalty weight, e.g. 1000

We show the parameter values used for each dataset in Table 11.

## P    EXPERIMENT ON HEART DISEASE

We evaluate our method on the real-world Heart Disease dataset from Kaggle[*]. This dataset contains records related to the diagnosis of heart disease in patients. Each record consists of features including patient demographics(e.g., age, gender), vital signs(e.g., resting electrocardiogram, resting heart rate, maximum heart) , symptoms(e.g., chest pain), and potential risk factors associated with heart

---

[*]https://www.kaggle.com/datasets/amirmahdiabbootalebi/heart-disease

conditions. Our target is to determine the presence or absence of heart disease in the patient. The Cholesterol value is taken as the continuous domains index, where the training dataset contains patients with Cholesterol value between (60.0, 220.0] and the testing dataset between (220.0, 421.0). The training dataset is equally split into discrete domains with 10 in each domain for existing methods dependent on discrete domains. Results in Table 12 show that all existing methods are inferior to ERM in terms of in-distribution training performance. However, all methods except IIBNet achieve a higher accuracy than ERM on OOD testing. Our CIL performs the best across all methods, improving by about 2% compared to other methods.

| Env. Type | Method | ID | OOD |
|---|---|---|---|
| None | ERM | 88.77(1.25) | 80.58(2.10) |
| Discrete | GroupDRO | 86.98(0.80) | 81.88(0.46) |
| | IIBNet | 81.64(0.55) | 77.67(1.21) |
| Continuous | IRMv1 | 87.24(2.07) | 83.17(1.65) |
| | REx | 87.76(1.87) | 82.85(0.92) |
| | Diversify | 87.24(1.21) | 82.52(2.86) |
| | EIIL | 87.36(0.48) | 82.13(1.25) |
| | HRM | 88.10(0.91) | 81.92(1.72) |
| | CIL | 86.23(1.25) | **84.79(0.92)** |

Table 12: Comparison on the HeartDisease datasets

## Q    ON THE CORRELATION OF $Y$

It is possible that the label is not independent of the domains. Taking the Heart Disease dataset as an example, we can visualize the proportion of positive labels among subgroups of patients with different Cholesterol values. The distribution of Y is shown as Figure 8. In this case, where the distribution of Y changes with the domain, we have observed that our method consistently outperforms ERM (Empirical Risk Minimization) and other competitive invariance learning methods. Additionally, we have explored another approach that involves re-weighting the samples to balance the Y ratio within each subgroup of the training data, where the subgroups are defined by Cholesterol intervals of 20.

For instance, let's consider a subgroup with a positive Y ratio of 0.33. We reweight the samples from this subgroup by a factor of 0.5/0.33. After the reweighting process, the Y ratio in each subgroup becomes 0.5. We have found that combining this reweighting technique with our CIL method achieves slightly better ID performance but slightly worse OOD performance. Notably, the OOD performance of reweighted CIL is still consistently better than existing methods when we compare Table 13 with Table 12.

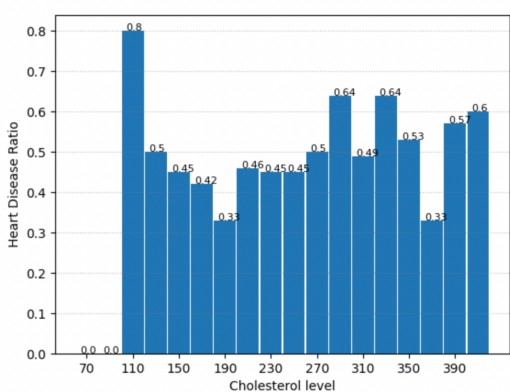

Figure 8: How the Y ratio changes with the Cholesterol value

| Method | ID | OOD |
|---|---|---|
| ERM | 88.77(1.25) | 80.58(2.10) |
| CIL | 86.23(1.25) | **84.79(0.92)** |
| CIL(re-weight) | **86.91(0.79)** | 84.12(1.36) |

Table 13: Comparison of re-weighted CIL with vanilla CIL

# R    ADDITIONAL EXPERIMENT ON CONTINUOUS CMNIST

The results in Table 2 in Section 4 shows a large variance Continuous CMNIST. We conjecture that it is due to that fact that invariance learning methods are prone to over-fitting in densely connected DNNs (Lin et al., 2022a; Zhou et al., 2022). To further improve and stabilize the performance of CIL on Continuous CMNIST, we try to incorporate methods in (Rosenfeld et al., 2022; Kirichenko et al., 2022).

We have conducted additional experiments on the Continuous CMNIST dataset. For feature extraction, we utilized a fixed pretrained ResNet18 model on the CMNIST images. In other words, we used the extracted features as inputs for our CIL. Our approach involved training only the linear layer on top of the fixed pre-trained features. This technique has been widely adopted in existing literature, where researchers have found that training just the last layer is sufficient because the pre-trained model already captures enough invariant and spurious features (Rosenfeld et al., 2022; Kirichenko et al., 2022). This method can significantly alleviate the overfitting issue. The results presented in Table 14 demonstrate that our CIL significantly benefits from being trained based on the fixed feature extracted by the pre-trained features, as they exhibit exceptional ID and OOD performance with low variance. Notably, the baseline methods are all trained on the feature extracted by a fixed pre-trained model and the other settings are the same with Section 4.1.1.

| Env. Type | Method | Linear | | | Sine | | |
|---|---|---|---|---|---|---|---|
| | | Split Num | ID | OOD | Split Num | ID | OOD |
| None | ERM | – | 84.84(0.01) | 10.60(0.08) | – | 85.17(0.01) | 10.58(0.18) |
| Discrete | IRMv1 | 8 | 75.68(0.77) | 52.06(1.18) | 2 | 76.20(0.15) | 52.35(0.45) |
| | REx | 4 | 78.42(0.73) | 39.30(4.00) | 4 | 70.19(0.03) | 62.22(0.20) |
| | GroupDRO | 2 | 84.73(0.01) | 12.04(0.27) | 16 | 85.00(0.01) | 12.28(0.15) |
| | IIBNet | 16 | 74.93(0.16) | 41.60(0.63) | 8 | 61.30(1.69) | 45.73(1.58) |
| Continuous | IRMv1 | – | 77.28(0.11) | 46.95(0.48) | – | 77.37(0.67) | 48.02(1.56) |
| | REx | – | 78.07(0.40) | 46.95(1.83) | – | 78.51(0.31) | 46.11(1.69) |
| | Diversify | – | 83.29(0.19) | 30.92(1.10) | – | 77.03(0.46) | 41.36(1.23) |
| | CIL | – | 70.33(0.63) | **62.15(1.37)** | – | 72.47(0.43) | **67.80(0.40)** |

Table 14: Accuracy on Continuous CMNIST for Linear and Sine $p_s(t)$. The standard deviation in brackets is calculated with 5 independent runs. The Env. type "Discrete" means that we manually create by equally splitting the raw continuous domains. The environment type "Continuous" indicates using the original continuous domain index. "Split Num" stands for the number of domains we manually create and we report the best performance among spilt $\{2, 4, 8, 16\}$.

