# OpenReview forum: "Continuous Invariance Learning"
_ICLR.cc/2024/Conference — ICLR 2024 poster_

### Official Review · Reviewer_9c4M · 2023-11-01

**Soundness:** 3 good
**Presentation:** 2 fair
**Contribution:** 3 good
**Rating:** 8
**Confidence:** 4

**Summary:**

The paper proposed Continuous Invariance Learning (CIL), a robust optimization algorithm targeting the continuous-domain setting. Through theoretical analysis, the authors show that existent methods fail to identify the spurious features when $\mathcal{T} \ge O(\sqrt{n})$, where $n$ is the sample size and $\mathcal{T}$ is the number of domains, whereas CIL can handle even the $\mathcal{T} \in O(n)$ case. Empirical experiments on both synthetic and real-world datasets indicate that CIL achieves SOTA performance.

**Strengths:**

CIL is motivated by a "duality" between domain and label due to the condition independence, which is an insightful observation. As a result, the method can target either continuous-domain discrete-label or discrete-domain continuous-label settings (the paper focuses on the former). It enjoys a strong theoretical guarantee and works well in practice, as indicated by the comprehensive experiments in which it outperforms all baselines.

**Weaknesses:**

While the method is titled "Continuous Invariance Learning", the theoretical analysis assumes countably many domains. For example, you can never have a domain with an irrational index like $\pi$ or $\sqrt{2}$. The authors may want to extend the theorems to cover the truly continuous setting. It is easy to see that the method itself applies to the truly continuous setting though.

According to Tables 3 and 4, CIL seems to perform poorly regarding in-domain prediction, with suboptimal accuracy and large variance. Moreover, I believe CIL has the potential to perform well in the discrete-domain continuous-label setting, but the authors didn't highlight or conduct experiments to demonstrate that.

Lesser issues:
1. Both $\mathcal{T}$ and $\mathcal{E}$ are used as the notation for domain indices, which is a little confusing.
2. In Table 5, you consider SWA as a baseline but didn't mention it in the main text.
3. The tables need some cleanup. In Table 4, you are claiming the accuracy for continuous IRMv1 under ID Alipay autoscaling is 885.7%. There are other issues like extra paratheses, missing space around paratheses, a missing period in the Table 3 caption, and an uncapitalized *method* in the Table 5 header.
4. In the paragraph starting with "Existing Approximation Methods": *Since is hard to validate* should be *Since it is hard to validate*.
5. In the paragraph just above subsection 4.2.4, *All The remaining data* should be *All the remaining data*.
6. In Appendix D.1: *Consider the following example, which has gained popularity* is repeated twice.

**Questions:**

What if I apply your method to a discrete-domain and discrete-label setting? How well does Appendix D.1's conclusion generalize to other cases? Does that depend on the relative cardinality of $\mathcal{Y}$ and $\mathcal{T}$?

In the introduction, you wrote *to regress over the continuous domain index $t$ using L1 or L2 loss*, but you only mention L2 loss in Section 3. What about the L1 loss?

Is $O(\sqrt{n})$ the slowest rate $|\mathcal{T}|$ can grow with respect to $n$ for REx/IRM to fail with a non-zero probability?

The proof in Appendix F is unclear: (i) the notations are not defined; (ii) the assumption that *each domain only contains one sample* contradicts *assuming we have infinite samples in each domain* in the main text; (iii) last sentence is confusing: did you mean *if $|\mathcal{E}| \ge \frac{\sigma_R \sqrt{n}}{\Delta \mathcal{G}^{-1}(1/4)}$ > 1/4, then REx can not identify the invariant feature with probability 1/4*?

---

> ### Author Response · Authors · 2023-11-19
> **Response 1/4**
>
> > Weakness 1:While the method is titled "Continuous Invariance Learning", the theoretical analysis assumes countably many domains. For example, you can never have a domain with an irrational index like $\pi$
>  or $\sqrt{2}$. The authors may want to extend the theorems to cover the truly continuous setting. It is easy to see that the method itself applies to the truly continuous setting though.
>
> Response:
> Thank you for your question. We proceed the response from three parts 1) Discussion on the theoretical analysis in Section 2; 2) Discussion on the theoretical analysis in Section 3; 3) Providing additional experimental results.
>
> **1)Discussion on the theoretical analysis in Section 2** In our theoretical analysis in Section 2, we initially consider countably many domains to align with existing methods. This approach allows us to highlight the limitations of those methods when there are multiple domains and each domain has a limited number of samples. However, our analysis can also encompass the truly continuous setting, where each sample has a continuous domain index, even if the domain index is irrational. In this case, the domain index of samples is typically different from each other, resulting in $n=|\mathcal{T}|$. The analysis in Section 2 still holds with the same conclusion: Proposition 2 shows that existing methods like REx fail to identify the invariant feature with a constant probability.
>
> **2) Discussion on the theoretical analysis in Section 3**. In Section 3.1, we analyze the strength of CIL. We do not assume that the domain index $t$ is countable. In fact, $t$ can be any continuous value, including irrational numbers. The only requirement is that the output variable $Y$ should be discrete and have limited cardinality, which is typically the case for classification tasks. Therefore, CIL can handle arbitrary continuous values for the domain index $t$.
>
> **3) Additional experimental results**. We have conducted additional experiments on a real-world Heart Disease dataset that consists of mechanically related domains, i.e., Cholesterol value. The Cholesterol value in this dataset is continuous and represented as floating-point numbers. This dataset contains records related to the diagnosis of heart disease in patients. Each record includes features such as patient demographics (e.g., age, gender), vital signs (e.g., resting electrocardiogram, resting heart rate, maximum heart rate), symptoms (e.g., chest pain), and potential risk factors associated with heart conditions. Our goal was to determine the presence or absence of heart disease in the patient. The Cholesterol value was used as the continuous domain index, with the training dataset containing patients with Cholesterol values between (60.0, 220.0] and the testing dataset between (220.0, 421.0). The results bellow demonstrate that our CIL consistently outperforms all baseline methods. We also add these experiments in  Appendix P of the revised version.
> | Env. Type   | Method     | ID           | OOD          |
> |-------------|------------|--------------|--------------|
> | None        | ERM        | 88.77(1.25)  | 80.58(2.10)  |
> | Discrete    | GroupDRO   | 86.98(0.80)  | 81.88(0.46)  |
> | Discrete    | IIBNet     | 81.64(0.55)  | 77.67(1.21)  |
> | Continuous  | IRMv1      | 87.24(2.07)  | 83.17(1.65)  |
> | Continuous  | REx        | 87.76(1.87)  | 82.85(0.92)  |
> | Continuous  | Diversify  | 87.24(1.21)  | 82.52(2.86)  |
> | Continuous  | EIIL       | 87.36(0.48)  | 82.13(1.25)  |
> | Continuous  | HRM        | 88.10(0.91)  | 81.92(1.72)  |
> | Continuous  | CIL        | 86.23(1.25)  | 84.79(0.92)  |
>
> If you have any further questions or need additional clarification, please let me know!

---

> ### Author Response · Authors · 2023-11-19
> **Response 2/4**
>
> > Weakness 2 (part1): According to Tables 3 and 4, CIL seems to perform poorly regarding in-domain prediction, with suboptimal accuracy and large variance.
>
> Response:
>
> We agree with you that the results for Continuous CMNIST in Table 3 is unsatisfactory.  We find that it might be due to the fact that invariance learning methods are prone to over-fitting in densely connected DNNs [1, 2]. [3, 4] has proposed effective methods to alleviate the overfitting issue. Specifically, they propose to use the feature extracted by a fixed pre-trained model as the input of the invariance learning algorithm. Invariance learning algorithms only train a linear layer on the top of the pre-trained feature. They have found that training just the last layer is sufficient for invariance learning because the pre-trained model already captures enough invariant and spurious features [3, 4]. We combine CIL with their methods by using a pre-trained ResNet18 as the fixed feature extractor and training only a linear layer.  The results presented bellow demonstrate that our CIL significantly benefits from being trained based on the fixed feature extracted by the pre-trained models. Take the Continuous CMNIST Linear dataset for example, the original CIL in Table 3 have an ID accuracy 57.35(6.98) and OOD accuracy 57.20(6.89); in contrast, the CIL with new implementation achieves ID accuracy 70.33(0.63) and OOD accuracy 62.15(1.37). We also re-implement the baseline methods by training them on the pretrained feature, which is the same with our CIL setting. The results bellow shows that CIL significantly outperform these baselines.
>
> | Env. Type | Method    | Split Num | (Continuous CMNIST Linear)ID            | (Continuous CMNIST Linear) OOD           | Split Num | (Continuous CMNIST Sine) ID            | (Continuous CMNIST Sine) OOD           |
> |-----------|-----------|-----------|---------------|---------------|-----------|---------------|---------------|
> | None      | ERM       | --        | 84.84(0.01)   | 10.60(0.08)   | --        | 85.17(0.01)   | 10.58(0.18)   |
> | Discrete  | IRMv1     | 8         | 75.68(0.77)   | 52.06(1.18)   | 2         | 76.20(0.15)   | 52.35(0.45)   |
> |           | REx       | 4         | 78.42(0.73)   | 39.30(4.00)   | 4         | 70.19(0.03)   | 62.22(0.20)   |
> |           | GroupDRO  | 2         | 84.73(0.01)   | 12.04(0.27)   | 16        | 85.00(0.01)   | 12.28(0.15)   |
> |           | IIBNet    | 16        | 74.93(0.16)   | 41.60(0.63)   | 8         | 61.30(1.69)   | 45.73(1.58)   |
> | Continuous| IRMv1     | --        | 77.28(0.11)   | 46.95(0.48)   | --        | 77.37(0.67)   | 48.02(1.56)   |
> |           | REx       | --        | 78.07(0.40)   | 46.95(1.83)   | --        | 78.51(0.31)   | 46.11(1.69)   |
> |           | Diversify | --        | 83.29(0.19)   | 30.92(1.10)   | --        | 77.03(0.46)   | 41.36(1.23)   |
> |           | CIL       | --        | 70.33(0.63)   | **62.15(1.37)** | --        | 72.47(0.43)   | **67.80(0.40)** |
>
> Table: Accuracy on Continuous CMNIST Linear and Sine.  The Env. type "Discrete'' means that we manually create by equally splitting the raw continuous domains. The environment type "Continuous'' indicates using the original continuous domain index. ``Split Num'' stands for the number of domains we manually create and we report the best performance among spilt (2, 4, 8, 16).
>
> [1] Yong Lin, et.al., Bayesian Invariant Risk Minimization
>
> [2] Xiao Zhou, et.al., Sparse Invariant Risk Minimization
>
> [3] Rosenfeld Elan, et.al., Domain-adjusted regression or: Erm may already learn features sufficient for out-of-distribution generalization
>
> [4] Kirichenko Polina, et.al., Last layer re-training is sufficient for robustness to spurious correlations

---

> ### Author Response · Authors · 2023-11-19
> **Response 3/4**
>
> > Weakness 2 (part2): Moreover, I believe CIL has the potential to perform well in the discrete-domain continuous-label setting, but the authors didn't highlight or conduct experiments to demonstrate that.
>
> Response:
>
> **Intuition**. We agree that our CIL can also be applied in the discrete-domain and continuous-label tasks. Whereas, existing methods can also achieve effective OOD performance on these tasks since the domains are discrete and we have sufficient samples in each domain. For example, we can also apply REx in these tasks by replacing their the cross entropy loss with L2 loss. The penalty of REx would simply be the variance of L2 loss in each domain.
>
> **Experiments**. We apply our method on the following variant of Alipy Auto-scaling task where we have continuous label with discrete domains.  We split continuous time domains (1440 minutes) into 4 discrete domains with 6 hours in each split, and our target is to predict the CPU utilization which is a real value in [0.0, 100.0].  Other settings are same as section 4.2.3. The table bellow shows the MSE of the prediction (the lower the better). Results show that our CIL can outperform ERM method in the OOD scenario. Meanwhile, existing methods such as REx and IRMv1 also achieve comparable performance with ours.
>
> |  Model   | ID  | OOD  |
> |  ----  | ----  |  ----  |
> | ERM  | 19.3(0.3) | 18.0(0.2) |
> | IRMv1  | 19.7(2.5) | 15.7(0.2)|
> | REx  | 19.6(4.7) | 16.2(2.1)|
> | CIL  | 18.9(0.5) | 15.9(0.2)|
>
> Table: MSE on the variant of Auto-scaling
>
> > Minor issues and typos
>
> Response
>
> Really sorry for the typos. Thank you for pointing them out. We have fixed the typos and carefully checked the paper over and over again. As for the domain indexes, we have unified them as $\mathcal{T}$ in the revised version. The SWA method is to adopt stochastic weight averaging for ERM. It is shown in [1] that SWA can improve the OOD performance. Since we adopt the same baseline methods with Wild-time and directly copy the performance of baseline methods from Wild-time, we kindly refer the readers to Appendix B of Wild-time for a detailed discriptions of each baselines.
>
> [1] Junbum Cha, et.al., SWAD: Domain Generalization by Seeking Flat Minima
>
>
> > Question 1 (Part 1): What if I apply your method to a discrete-domain and discrete-label setting?
>
> Response:
>
> We can apply CIL to discrete-domain and discrete-label setting. For example, we  try our CIL on the original CMNIST in [1] with two discrete domains for binary classification. In this case, our method achieves similar performance with existings methods such as REx and IRMv1. The results are summarized as follows:
>
> | Method | ID   | OOD  |
> |--------|------|------|
> | ERM    | 87.4 | 17.1 |
> | IRMv1  | 70.8 | 66.9 |
> | REx    | 69.8 | 67.5 |
> | CIL    | 70.2 | 67.2 |
>
> Table: Accuracy of CIL on CMNIST with discrete-domain and discrete-label.
>
> > Question 1 (Part 2):  How well does Appendix D.1's conclusion generalize to other cases? Does that depend on the relative cardinality of Y and T ?
>
> **Assuming infinite samples in each domain and class**. The analysis in Appendix D.1 suggests that, for a problem with two classes and two discrete environments, if we have an infinite number of samples for each class and environment, we can identify the invariant feature by matching either $E^y[t|x]$ or $E^t[y|x]$. It becomes interesting when we consider increasing the cardinality of $\mathcal{T}$ or $\mathcal{Y}$. When the number of domains increases, assuming we still have an infinite number of samples in each domain (although this assumption is realistic), we can easily identify the invariant features by matching $E^t[y|x]$ in each domain. For example, we can identify a spurious feature $x_s$ as long as there exists two domains $t_1$ and $t_2$ such that $E^{t_1}[y|x_s] \neq E^{t_2}[y|x_s]$.
>
> **More realistic situations**. However, let's consider a more realistic scenario in real-world applications where we are given a collected dataset. If the samples are distributed among many domains, then the sample size in each domain would be rather limited. The results in Proposition 1 shows that matching the empirical $\hat E^t[y|x_s]$ in each domain can be problematic due to estimation errors.
>
> > Question 2: In the introduction, you wrote to regress over the continuous domain index using L1 or L2 loss, but you only mention L2 loss in Section 3. What about the L1 loss?
>
> Response:
>
> Thank you for the question. We apologize for not including experiments with L1 loss initially. We have conducted additional experiments using L1 loss on three datasets: HousePrice, Insurance Fraud, and Alipay Auto-scaling. The results are presented in Table 4 of the revised version. "CIL(L1)" refers to using L1 loss in our CIL method, while "CIL(L2)" represents the original version with L2 loss. We can observe that CIL(L1) performs comparably to CIL(L2). Both CIL(L1) and CIL(L2) consistently outperform existing methods.

---

> ### Author Response · Authors · 2023-11-19
> **Response 4/4**
>
> > Question 3: Is $O(\sqrt{n})$ the slowest rate $T$ can grow with respect to $n$ for REx/IRM to fail with a non-zero probability?
>
> Response:
>
> Yes. The rate $O(\sqrt{n})$ is asymptotically optimal. For example, if $n$ and $\mathcal{T}$ both goes to infinity, we could see $O(\sqrt{n})$ should be exactly the order.  Specifically, assume  $n$ and $\mathcal{T}$ both go to infinity, according to Appendix F,  we have $\hat L(\Phi_v) - \hat L(\Phi_s)$ converges to $- \sigma_R |\mathcal{T}|+ \frac{\sigma_v - \sigma_s}{\sqrt{n}}|\mathcal{T}|^2$.  Since $\sigma_R$, and $\sigma_v - \sigma_s$ is in constant level, we can see if the REx penalty fails when $|\mathcal{T}|$ equals or is larger than $\sqrt{n}$. If $|\mathcal{T}|$ is in slower rate than $\sqrt{n}$ , REx won't fail with a constant probability.
>
> > Question 4: The proof in Appendix F is unclear: (i) the notations are not defined; (ii) the assumption that each domain only contains one sample contradicts assuming we have infinite samples in each domain in the main text; (iii) last sentence is confusing: did you mean if ... then REx can not identify the invariant feature with probability 1/4?
>
> Response:
>
> Thank you for your valuable questions.
>
> (i) We apologize for missing the notations in the appendix. We have updated the appendix by adding the necessary annotations for clarity and improved readability. Thank you for bringing this to our attention.
>
> (ii) Apologies for the confusion. In the "Remark on the Setting and Assumption" section, we discuss two cases. Case (1) assumes an infinite number of samples in each domain, where REx can successfully identify the invariant feature. Case (2) considers the scenario with multiple domains and limited samples in each domain, where REx may fail to identify the invariant feature with a certain probability. We present these cases to highlight the challenges posed by "many domains and limited sample sizes." Proposition 1 and the proof in Appendix F support the claim in Case (2). We apologize for any confusion caused and have revised our manuscript to clarify these points.
>
> (iii) You are correct. The statement "If  $|\mathcal{T}| \geq \frac{\sigma_R\sqrt{n}}{\Delta\mathcal{G}^{-1}(1/4)},$
> REx is unable to identify the invariant feature with a probability of at least 1/4" is accurately interpreted. Sorry that there is a typo that $\mathcal{E}$ should be $\mathcal{T}$, which is the space of domains. We have made the necessary updates to the manuscript to reflect this. Thank you for your suggestions.

---

> ### Comment · Area_Chair_PEQq · 2023-11-20
> **please add some comments on the rebuttal**
>
> Dear reviewer 9c4M,
>
> this paper is still in the "borderline" region, and in order to come to a conclusion, it would be very helpful to see your comments about the other reviews and the rebuttal.
>
> Thanks again,
> AC

---

> ### Author Response · Authors · 2023-11-22
>
> Dear review 9c4M:
>
> Thank you for your time and effort in reviewing our paper.
>
> We firmly believe  that our response and revisions can fully address your concerns. We are open to discussion if you have any additional questions or concerns, and if not, we kindly ask you to reevaluate your score.
>
> Thank you again for your reviews which helped to improve our paper!
>
> Authors

---

> > ### Comment · Reviewer_9c4M · 2023-11-23
> >
> > Thanks for the additional clarification, analysis, ablation, and experiments. Since the authors addressed most of my concerns, I am raising my rating. In particular, the result in "Table: MSE on the variant of Auto-scaling" looks promising: CIL has a much smaller variance compared to other methods.
> >
> > To provide an additional reason for acceptance: their method has been battle-tested in the industry ("it has been applied on Alipay Cloud to achieve the auto-scaling of server resources"). I'm glad the authors are making an actual impact with their expertise, and we should reward their action of bridging academia and industry.

---

> ### Author Response · Authors · 2023-11-23
>
> Dear Review 9c4M:
>
> Thank you so much for raising the score and your strong support. We are really glad that our response can address your concerns.

---

### Official Review · Reviewer_dGY4 · 2023-11-01

**Soundness:** 3 good
**Presentation:** 3 good
**Contribution:** 2 fair
**Rating:** 5
**Confidence:** 4

**Summary:**

This paper extends the invariant risk minimization (IRM) from discrete environment index to continuous environment index, in which there is no explicit environment partition and the domain variable is continuous. Authors first identify that some typical invariant representation learning algorithms, such as REx and IRMv1, may fail when the number of environments is large and there are only limited samples in each environment. Authors then propose a new regularization term by making $y$ is independent to $t$ (the domain index) given the extracted features $\Phi(X)$, and uses a min-max scheme to approximate the degree of independence by training two regression functions to fit
$p(t|\Phi(X))$ and $p(t|\Phi(X),y)$.

**Strengths:**

1. The movation of extending invariant representation learning from discrete environment to continuous environment is nice and useful.
2. Applications on Alipay and Wilds-time demonstrate the practical usages of the new method.

**Weaknesses:**

Frankly speaking, I really like the motivation of this paper. However, I just feel the main strategy (especially the way to measure conditional independence) is not novel to me, which has been used in previous extensions of IRM, such as InvRat (Chang et al., 2020) and IIBNet (Li et al., 2022). Moreover, the proof to Proposition 2 seems largely rely on result of (Li & Liu, 2021), including their assumptions.

1. The estimation on the degree of independence between y and t given $\Phi(X)$ does not seem novel to me. In fact, for discrete environment index $e$, both InvRat (Chang et al., 2020) and IIBNet (Li et al., 2022) use the same strategy to measure
$I(y;e|\Phi(x))$ ($I$ is the conditional mutual information) by minimizing the maximum difference of two prediction losses: 1) use $\Phi(X)$ to predict $y$; 2) use  $(\Phi(X),e)$ to predict $y$.

The only difference is that this paper changes the role of $y$ and $e$ and estimate $I(e;y|\Phi(x))$ by minimizing the maximum difference of two new prediction losses: 1) use $\Phi(X)$ to predict $e$; 2) use  $(\Phi(X),y)$ to predict $e$.

Given the fact that the conditional mutual information is symmetric, i.e., $I(y;e|\Phi(X))=I(e;y|\Phi(X))$, and it is equivalent to write this condition with either $p(y|\Phi(X))=p(y|\Phi(X),e)$ or $p(e|\Phi(X))=p(e|\Phi(X),y)$, it does not make a big difference or novelty to change the prediction target. Although I admit that in continuous regime, learning two functions to predict e makes more sense.

2.  I would like to see some comparisons with environment partition approaches (see below), in which there is no explicit discrete environment index. In principle, those approaches should perform slighly better than manually environment splitting.

[1] Creager, Elliot, Jörn-Henrik Jacobsen, and Richard Zemel. "Environment inference for invariant learning." International Conference on Machine Learning. PMLR, 2021.

[2] Liu, Jiashuo, et al. "Heterogeneous risk minimization." International Conference on Machine Learning. PMLR, 2021.

Some other points:
1. Section I should be proof to Proposition 2, rather than Theorem 2?
2. How to manually split environments for IRMv1, IIBNet, etc? By clustering? or did I miss something in the supplementary material?

**Questions:**

Please see above weaknesses point 2 and minor point 2.

---

> ### Author Response · Authors · 2023-11-19
> **Response 1/2**
>
> > Weakness 1: ``Frankly speaking, ..., make more sense.''
>
> Response:
>
> **Main Intuition**. We are really delighted that you like the motivation of our paper. We acknowledge that the InvRat and IIBNet also minimizes the mutual information $I(y; e|\Phi(x))$ by training two estimators. But as you have mentioned, our parameterization to predict $e$ is more compatible with continuous domain. Specifically, we use  L2 distance between the predicted and ground-truth domains to capture the underlying distance between domain indices. Existing works also show that when the domains index become more complex, InvRat would be less effective [1]. For example, [1] shows that if we increase the domain number from 2 to 4 on the CIFARMNIST dataset and keep the other settings fixed, the OOD performance of InvRat drops from 77.6±2.0 to 68.6±9.4.
>
> **Empirical Results** We have shown in the experiments part, the performance of our CIL outperform IIBNet consistently in all the benchmarks, indicating the effectiveness of our parameterization methods. We also conduct additional experiments with InvRat on HousePrice, Insurance Fraud and Alipay Auto-scaling. The results are shown in Table 3 of the revised version. We can see that our CIL can also consistently outperform InvRat.
>
> **Clarifications on the Proof of Proposition 2 and our main theoretical contribution**. First we would like to clarify our main theoretical contribution is to show the disadvantage of existing methods in continuous domain tasks (i.e., Proposition 1) and the advantage our CIL (i.e., Theorem 2). For example, consider a 10-class classification task with an infinite number of domains and each domain contains only one sample, i.e., $n \xrightarrow[]{} \infty$, and $n / |\mathcal{T}| = 1$, Proposition 1 shows that REx would fail to identify the invariant feature with constant probability. Whereas, Theorem 2 shows that CIL can still effectively extract invariant features. Details are in Section 3.1. We already showed that the definite advantage of CIL merely based on Proposition 1 and Theorem 2, even without involving Proposition 2. Proposition 2 is added to show the finite sample property of our CIL for completeness. As you mentioned, Proposition 2 is a standard analysis of mini-max formulation and we use the results of (Li and Liu, 2021). We have added discussions on this part to avoid confusion. Thank you for your suggestion!
>
> [1]   Yong Lin, et.al., An empirical study of invariant risk minimization on deep models
>
> > Weakness 2: I would like to see some comparisons with environment partition approaches (see below), in which there is no explicit discrete environment index. In principle, those approaches should perform slighly better than manually environment splitting.
>
> Response:
>
> We have conducted additional experiments on methods that do not require manual partitions, such as EIIL and HRM. The results of these experiments are included in Table 4 of the revised version. Our methods continue to consistently outperform EIIL and HRM. It is worth noting that we observed instability in EIIL and HRM. Specifically, in the HousePrice and Insurance datasets shown in Table 4, both EIIL and HRM show improved performance compared to other baselines. However, when it comes to Alipay Auto-scaling, HRM and EIIL are not effective. The instability of HRM and EIIL could be due to identifiability issues mentioned in reference [1]. Specifically, [1] shows that it is generally impossible to learn invariance without domain partition. HRM and EIIL rely on implicit inductive bias to achieve domain partition. However, such inductive bias can fail in many real-world tasks.
>
> We also explored an alternative method called ZIN [1], which addresses the identifiability issue by using our continuous domain as the auxiliary information. ZIN proves to be more stable than HRM and EIIL. However, our method still consistently outperforms ZIN in terms of performance.
>
> [1] Yong Lin, et.al., ZIN: when and how to learn invariance without domain partition.

---

> ### Author Response · Authors · 2023-11-19
> **Response 2/2**
>
> > minor point 1: Section I should be proof to Proposition 2, rather than Theorem 2?
>
> Response:
>
> Thank you for pointing out these typos! We have fixed them and checked our manuscript over and over again.
>
> > minor point 2: How to manually split environments for IRMv1, IIBNet, etc? By clustering? or did I miss something in the supplementary material?
>
> Response:
>
> We experimented with different split schemes for the baseline methods. Firstly, we tested them on the vanilla datasets without any manual splitting. Secondly, we also tried these baselines on manually split settings where we equally divided the domain into different numbers of discrete domains. We determined the best manual domain split for each baseline and reported the best performance among the manually split schemes. For example, in the Logit and CMNIST experiment, we tried the baseline methods on an equal split of 2, 4, 8, and 16 domains, and selected the domain split that resulted in the best OOD performance for each method. Take the logit dataset for example, the domain index is $t \in [0, 100]$, "equally splitting into 2 environments" means that we manually assigns the sample in $t \in [0, 50)$ to one environment and the rest samples to another environment. Additionally, in the House Price and Insurance Fraud datasets, we tried  splitting the years into 5 or 10-year intervals for baselines. In the Cloud dataset, we split it by hour. More details can be found in Appendix L and M where we have provided further elaboration on this topic.

---

> ### Author Response · Authors · 2023-11-22
>
> Dear Reviewer dGY4:
>
> Thank you for your time and effort in reviewing our paper.
>
> We firmly believe  that our response and revisions can fully address your concerns. We are open to discussion if you have any additional questions or concerns, and if not, we kindly ask you to reevaluate your score.
>
> Thank you again for your reviews which helped to improve our paper!
>
> Authors

---

### Official Review · Reviewer_MYYo · 2023-11-02

**Soundness:** 4 excellent
**Presentation:** 4 excellent
**Contribution:** 3 good
**Rating:** 8
**Confidence:** 4

**Summary:**

This paper addresses the problem of generalization in continuous domains. Specifically, they aim to learn invariant features in settings characterized by continuous domain. Previous methods are restricted to work with the discrete indexed domains.

The authors demonstrate theoretically that when the number of domains is large and the number of samples from each domain is finite, existing methods can fail. Thus, discretizing continuous domains may not be a good solution. The authors then propose a min-max objective to learn invariant features by aligning domain distribution for each class of given features. To align the distributions, they use two domain predictors --- one uses invariant features, and the other uses label and invariant features. The goal is to learn features such that knowledge of labels does not increase the information about the domain.

Authors tested their method on several toy and real-world datasets with continuous domains. Their approach outperforms in all the cases.

**Strengths:**

- The arguments in the paper flow well, and the problem formulation is interesting. This paper could be a benchmark (for datasets & settings) for future work exploring generalization over continuous domains.
- Authors provide theoretical evidence to explain the failure of existing methods. This complements the empirical results that demonstrate the same.
- The authors demonstrate the superior performance of their method across various real-world and toy datasets

**Weaknesses:**

- In almost all the datasets considered in the paper, the ground truth labels are independent of domains. However, it is possible to have domains where these are correlated --- different amounts of correlation in different domains. Why was such a toy environment not considered? It would be interesting to see how the proposed method performs in more correlated settings.

- The proposed approach relies on classes being discrete, whereas the prior method relies on the domain being discrete. This limitation should be highlighted in the paper.


-------
Minor Typo Issues:
- Incorrect use of braces for citations (not an exhaustive list; please check the paper carefully):
  - Sec 1. First para: use bracketed citations for He et al. (2016) etc,
  - Sec 2. .. following (Arjovsky et al., 2019) ...
- Sec 1, Para 3 last line, incorrect brackets
- Sec 2: Invariance learning Para: ... to extracted Φ(x) ... -> to extract
- Sec 2: ...variance of the losses in among domains -> ...variance of losses among domains.
- Sec 3: Formulation para: Shouldn't soft regularization eqn have "- t" in the last two terms?
- Tables 3 and 4: Some values are too high in ID columns

**Questions:**

## Hyperparameters
- What is the size of invariant features, i,e., $\phi (x)$? Is the method sensitive to the size of these features?
- What NN architecture & hyperparameters were used for training?


## On linear scaling of penalty
- In the "Remark on the settings and assumption" paragraph, how does the penalty scale linearly with the number of environments when the spurious mask is used? We have assumed $\mathcal R^t(w, \phi) \sim \mathcal N (\mathcal R(w, \phi), \delta_R)$. So, shouldn't the variance penalty be $\delta_R$ when the spurious mask is used?

---

> ### Author Response · Authors · 2023-11-19
> **Response 1/2**
>
> > Weakness 1: In almost all the datasets considered in the paper, the ground truth labels are independent of domains. However, it is possible to have domains where these are correlated --- different amounts of correlation in different domains. Why was such a toy environment not considered? It would be interesting to see how the proposed method performs in more correlated settings.
>
> Response:
>
> It is possible that the label is not independent of the domains. For example, we conduct an additional experiment on the Heart Disease dataset (binary classification task) in Appendix P where the Cholesterol value is the domain. We visualize the proportion of positive labels among subgroups of patients with different Cholesterol values. The distribution of Y is shown as Figure 6 in Appendix Q. In this case, where the distribution of Y changes with the domain, we have observed that our method consistently outperforms ERM and other competitive invariance learning methods. Additionally, we have explored another approach that involves re-weighting the samples to balance the Y ratio within each subgroup of the training data, where the subgroups are defined by Cholesterol intervals of 20. For instance, let's consider a subgroup with a positive Y ratio of 0.33. We reweight the samples from this subgroup by a factor of 0.5/0.33. After the reweighting process, the Y ratio in each subgroup becomes 0.5. We have found that combining this reweighting technique with our CIL  method achieves slightly better ID performance but slightly worse OOD  performance. Notably, the OOD performance of reweighted CIL is still consistently better than existing methods.
>
> | Env. Type   | Method     | ID           | OOD          |
> |-------------|------------|--------------|--------------|
> | None        | ERM        | 88.77(1.25)  | 80.58(2.10)  |
> | Discrete    | GroupDRO   | 86.98(0.80)  | 81.88(0.46)  |
> | Discrete    | IIBNet     | 81.64(0.55)  | 77.67(1.21)  |
> | Continuous  | IRMv1      | 87.24(2.07)  | 83.17(1.65)  |
> | Continuous  | REx        | 87.76(1.87)  | 82.85(0.92)  |
> | Continuous  | Diversify  | 87.24(1.21)  | 82.52(2.86)  |
> | Continuous  | EIIL       | 87.36(0.48)  | 82.13(1.25)  |
> | Continuous  | HRM        | 88.10(0.91)  | 81.92(1.72)  |
> | Continuous  | CIL        | 86.23(1.25)  | 84.79(0.92)  |
> | Continuous  | CIL(Reweight)        | 86.91(0.79) | 84.12(1.36)  |
>
> Table: Performance on Heart Disease. "Continuous" indicates that we use the original Cholesterol values as domains. "Discrete" means that we manually split the dataset into discrete domains by Cholesterol values with an interval of 10. This is to accommodate for existing methods that is designed for discrete domains.
>
>
> > Weakness 2: The proposed approach relies on classes being discrete, whereas the prior method relies on the domain being discrete. This limitation should be highlighted in the paper.
>
> Response:
>
> Thank you for the valuable suggestion! We completely agree with you. Our method is based on the assumption that the class variable is discrete. Notably, the current applications of invariance learning methods primarily focus on classification tasks, where the class variable is typically  discrete. So our methods are applicable in these tasks. Previous methods assume that the domains are discrete, which is not applicable in many applications with continuous domains. We have added this part in the Section 3 of the revised paper.
>
>
> > Minor points, Typos
>
> Response:
>
> Really sorry for the typos. Thank you for pointing them out. We have fixed the typos and carefully checked the paper over and over again. The term in soft regularization formulation in Section 3 should be $\|h(\Phi(x)) - t\|$ instead of $\|h(\Phi(x)), t\|$.  We also fixed the decimal point issue in Table 3 and 4.

---

> ### Author Response · Authors · 2023-11-19
> **Response 2/2**
>
> > Question 1: What is the size of invariant features, i,e., $\Phi(x)$? Is the method sensitive to the size of these features?
>
> Response:
>
> **The dimension of $\Phi(x)$**. Thanks for your question. In our study on Alipay Auto-scaling, we set the dimension of $\Phi(x)$ to be 128.  We further conduct experiments by changing the dimension of $\Phi(x)$ to 32, 64, 256, and 512, and found that the performance of our method remained stable when varying this hyper-parameter, as shown in Table 10.
>
> **Discussion on the number of invariant features**. It is important to note that the dimension of $\Phi(x)$ does not necessarily correspond to the number of invariant features.  In a previous study [1], it was demonstrated that it is even possible to set the dimension of $\Phi(x)$ to 1. In this case, the entire neural network outputs a one-dimensional $\Phi(x)$, which is then multiplied by another scalar $w$ to serve as the classifier. Despite being one-dimensional, $\Phi(x)$ can still be composed of multiple invariant features. Therefore, the dimension of $\Phi(x)$ is not necessarily linked to the number of spurious features.
>
>
> > Question 2: What NN architecture and hyperparameters were used for training?
>
> Response:
>
> For the Logit experiment (Section 4.1.1), we used a linear layer. In the CMNIST experiment (4.1.2), we employed a three-layer MLP with a hidden size of 390, which is consistent with previous research [1]. For the real-world datasets (Section 4.2), we utilized a two-layer MLP with hidden dimensions of 128. Detailed hyper-parameters can be found in Appendix O.
>
>
> > Question 3: In the "Remark on the settings and assumption" paragraph, how does the penalty scale linearly with the number of environments when the spurious mask is used? We have assumed . So, shouldn't the variance penalty be  when the spurious mask is used?
>
> Response:
>
> Sorry we have a typo in the formulation in Equation (1) of Section 2.1. The REx loss we use should be $L = \sum_{t \in \mathcal{T}} \mathcal{R}^t(w, \Phi) + \lambda |\mathcal{T}|\text{Var}(\mathcal{R}^t(w, \Phi))$. The penalty grows linearly with the domain number. We re-scale the original REx loss [2], i.e.,  $L = \frac{1}{|\mathcal{T}|}\sum_{t \in \mathcal{T}} \mathcal{R}^t(w, \Phi) + \lambda \text{Var}(\mathcal{R}^t(w, \Phi))$, by multiplying with a factor of $|\mathcal{T}|$ for ease of presentation. However, such re-scaling would not change the relative relationship between  $L(\Phi_s)$ and $L(\Phi_v)$, thus would not change the analysis results. We have fixed this typo and added a corresponding remark in Section 2.1. Thank you for pointing out this problem.
>
> [1] Martin Arjovsky, et al., Invariant Risk Minimization.
>
> [2] David Krueger, et.al., Out-of-distribution generalization via risk extrapolation

---

### Official Review · Reviewer_Zzch · 2023-11-08

**Soundness:** 3 good
**Presentation:** 2 fair
**Contribution:** 3 good
**Rating:** 5
**Confidence:** 3

**Summary:**

This paper proposed the Continuous Invariance Learning(CIL) method that extends invariance learning from discrete categorical indexed domains to continuous domain in this paper.  Empirical results on both synthetic and real-world tasks demonstrated that CIL achieves improvements over baselines.

**Strengths:**

investigating invariant learning for continuous domains is an interesting idea. This paper presents some innovative contributions and can extend the invariance learning among discrete domains to continuous domains, which has certain reference significance for other studies. As a theoretical rooted work, it first proves that the existing method fails in the continuous domain through theoretical derivation, and then proves the effectiveness of its own method. Also, the empirical studies verify the effectiveness of the work.

**Weaknesses:**

Although the contributions of this work is worth noting, it still has some limitations in terms of problem definition and presentation. First is about the problem setting: in my view, when we talk about *continuous* in machine learning, it will reflect some time-series issues, i.e., the continual learning or lifelong learning framework. However, it seems in this work the notion of *continuous* is related to *many* domains. It’s more like we have several intermediate domains between two discrete domains. I’m wondering whether the notion of *continuous* is accurate here.
Second is about the presentation of the paper. Some fonts of the tables and figures are to small, making them hard to read. This article can be further optimized in terms of visualization. Besides, there are also many grammar errors, which makes the paper less readable, e.g., under Eq. 1, “…in among…”, in the subsection of *theoretical analysis of REx*, ‘Eq. equation 1’

**Questions:**

1. The real-world datasets implemented in this paper, such as the HousepPrice dataset, take the built year as the continuous domain index. While in real life, adjacent years are separated by 1 year. Can this be regarded as a continuous domain? Same question with the Insurance Fraud datasets.

2. In the paragraph *Existing Approximation Methods*, the authors let wt denote the optimal classifier for domain *t*. For the continuous domains, are there are continuous and infinite classifiers for each domain? How does such an assumption work in practice?

3. In the paragraph ‘Formulation’, the authors say that since if $y ⊥ t|Φ(x)$, the loss achieved by $g(Φ(x),y)$ would be similar to the loss achieved by h(Φ(x)). If the loss of $g(Φ(x),y)$ is similar to the loss of  $h(Φ(x))$, then will $y ⊥ t|Φ(x)$ hold？

4. In the paragraph ‘Empirical Verification’, the samples are simulated to distribute uniformly on the domain index $t$. According to figure 5, there is a step up of the ps(t). Please explain the reason.

5. As for the experiments on real-world dataset, I am wondering if it is more convincing to expand the verification on some mechanically related data sets. Because we usually think of information such as speed and speed as continuous rather than discrete.

---

> ### Author Response · Authors · 2023-11-19
> **Response 1/2**
>
> Since the following three questions are highly related, we reponse to them togethor:
>
> > Weakness: Although the contributions of this work is worth noting, it still has ... notion of continuous is accurate here.
>
> > Question 1: The real-world datasets implemented in this paper, such as the HousepPrice dataset,... separated by 1 year. Can this be regarded as a continuous domain? Same question with the Insurance Fraud datasets.
>
> > Question 5: As for the experiments on real-world dataset, I am wondering if it is more convincing to expand the verification on some mechanically related data sets. Because we usually think of information such as speed and speed as continuous rather than discrete.
>
> Response: Thanks for pointing out the valuable question. We will address it by four steps: 1. The original meaning of continuous domain; 2. Natural discretization when recording data; 3. Consideration of a domain as continuous; and 4. Additional experiments on mechanically related domains.
>
>  **1. The original meaning of continuous domain**: A continuous domain refers to a situation where the domain index is associated with a continuous variable. For instance, this continuous variable could represent time slots, speed, BMI index, or cholesterol levels in the blood. In such cases, there can be an infinite number of domains, and the domain values can even be irrational. Importantly, the domain values can capture the distance between different domains. For example, in the case of time, if we have domain values $t_0$, $t_1$, and $t_2$, where $t_0 < t_1 < t_2$, then the domain $t_1$ is closer to $t_0$ compared with $t_2$ to $t_0$. When we collect a dataset containing samples from such domains, it is highly unlikely that two samples would share the same domain index. Our theoretical results (Proposition 1) demonstrate that existing methods can easily fail in these cases.
>
> **2. Natural discretization when recording data**: Although many tasks involve continuous domain values, it is common for humans to perform some form of discretization when recording the data. For example, even if a time index could be irrational, like $t=\sqrt{2}$ or $t=\pi$, we typically record the time as a rounded value, such as using $1.414$ for $\sqrt{2}$, keeping the last three decimal places. This discretizes the continuous domain value according to a specific interval, such as 0.001. A similar case can be seen with the year in the Houseprice dataset, where the time of house sale, although naturally continuous, is discretized by yearly intervals for ease of recording and visualization. In the insurance fraud dataset, the time since a person's birth is continuous, but it is discretized into one-year intervals to represent their age.
>
> **3. Consideration of a domain as continuous**: Despite the discretization that often occurs during data collection for continuous domains, we argue that they can still be valid datasets for validating our methods if they meet the following two criteria: (1) the domain index implies a distance measure; it can capture the intrinsic relationships between different domains, e.g., the year 1995 being closer to 1990 than 2000 to 1990, and (2) there is a considerable number of domains with only a limited number of samples in each domain. If the domain indices satisfy these two criteria, conventional invariance learning methods would fail, whereas our method would perform better. Notably, these two criteria also encompass the "real" continuous domain where no discretization is performed. First the "real" continuous domain index indicates the relationship between domains (which satisfies criterion 1). Second, for "real" continuous domain, each sample typically has different domain index from another sample. Then in a dataset with such "real" continuous domain, we have a great number of domains while each domain probably contains one data, satisfying criterion 2.
>
> **4. Additional experiments on mechanically related domains**: We conducted additional experiments on a real-world Heart Disease dataset. This dataset contains records related to the diagnosis of heart disease in patients. Each record includes features such as patient demographics (e.g., age, gender), vital signs (e.g., resting electrocardiogram, resting heart rate, maximum heart rate), symptoms (e.g., chest pain), and potential risk factors associated with heart conditions. Our goal was to determine the presence or absence of heart disease in the patient. The Cholesterol value, a real number,  was used as the continuous domain index, with the training dataset containing patients with Cholesterol values between (60.0, 220.0] and the testing dataset between (220.0, 421.0). The results bellow demonstrate that our CIL consistently outperforms all baseline methods. We also add these experiments in  Appendix P of the revised version.

---

> ### Author Response · Authors · 2023-11-19
> **The additional experimental results of Response 1/2**
>
> | Env. Type   | Method     | ID           | OOD          |
> |-------------|------------|--------------|--------------|
> | None        | ERM        | 88.77(1.25)  | 80.58(2.10)  |
> | Discrete    | GroupDRO   | 86.98(0.80)  | 81.88(0.46)  |
> | Discrete    | IIBNet     | 81.64(0.55)  | 77.67(1.21)  |
> | Continuous  | IRMv1      | 87.24(2.07)  | 83.17(1.65)  |
> | Continuous  | REx        | 87.76(1.87)  | 82.85(0.92)  |
> | Continuous  | Diversify  | 87.24(1.21)  | 82.52(2.86)  |
> | Continuous  | EIIL       | 87.36(0.48)  | 82.13(1.25)  |
> | Continuous  | HRM        | 88.10(0.91)  | 81.92(1.72)  |
> | Continuous  | CIL        | 86.23(1.25)  | 84.79(0.92)  |
> Table: Performance on Heart Disease. "Continuous" indicates that we use the original Cholesterol values as domains. "Discrete" means that we manually split the dataset into discrete domains by Cholesterol values with an interval of 10. This is to accommodate for existing methods that is designed for discrete domains.
> If you have any further questions or need additional clarification, please let me know!

---

> ### Author Response · Authors · 2023-11-19
> **Response 2/2**
>
> > Weakness: Second is about the presentation of the paper. Some fonts of the tables and figures are to small, making them hard to read. This article can be further optimized in terms of visualization. Besides, there are also many grammar errors, which makes the paper less readable, e.g., under Eq. 1, “…in among…”, in the subsection of theoretical analysis of REx, ‘Eq. equation 1’.
>
> Response:
>
> Sorry for the typos and issues of illustrations. We have fixed the typos and carefully checked the paper over and over again. As for the size of table and figure size, we will try to fix them in the final version. We plan to put some tables, e.g., Table 3, in appendix and then we will have more space to fit other tables in the main part.
>
> > Question 2: In the paragraph Existing Approximation Methods, the authors let wt denote the optimal classifier for domain t. For the continuous domains, are there are continuous and infinite classifiers for each domain? How does such an assumption work in practice?
>
> Response:
>
> **The ideal method**. In the paragraph "Existing Approximation Methods", we first introduce existing methods which usually assume the domain index is categorical. Since they assume categorical domain index and each domain contains sufficient number of samples, they can ideally solve $w^t$ in this case. However, if we consider continuous domain where each sample could belong to different domain index, estimating $w^t$ for each $t$ could be extremely difficult and sometimes even not practical, which is also our main argument in Proposition 1.
>
> **The approximate methods**. Notably, though existing method share the same motivation that try to solve the optimal $w^t$ for each domain, they usually use approximation methods and try to avoid explicitly solving $w^t$. For example, REx calculate, $\mathbb{V}[\hat R^t(w, \Phi)]$, which is the variance of the empirical loss $\hat R^t(w, \Phi)$ in each domain. Though we can still calculate $\mathbb{V}[\hat R^t(w, \Phi)]$ for a task with continuous domain, Proposition 1 shows that $\mathbb{V}[\hat R^t(w, \Phi)]$ would differ significantly from the variance of expected loss in each domain $\mathbb{V}[R^t(w, \Phi)]$ in this case, leading to the failure of identifying invariant features .
>
>
> > Question 3: In the paragraph ‘Formulation’, the authors say that since .... hold?
>
> Response: Typically it is hard to directly verify the exact $y \perp t | \Phi(x)$. Existing methods try to align $\mathbb{E}^t[y|\Phi(x)]$ for different domains $t$. Our method try to align $\mathbb{E}^y[t|\Phi(x)]$ for different domains $y$. In other words, if  $g(\Phi(x), y)$ is similar to the loss of $h(\Phi(x))$, we would have $\mathbb{E}^y[t|\Phi(x)] = \mathbb{E}[t|\Phi(x)]$ for each $y$, which means we match the first moment of $y \perp t | \Phi(x)$. Notably, to the best of our knowledge, nearly all existing methods in invariance learning all try to approximate $y \perp t | \Phi(x)$ by matching the first order moment. It is interesting to further explore the higher order moment matching in the future work.
>
> > Question 4: In the paragraph ‘Empirical Verification’, the samples are simulated to distribute uniformly on the domain index t. According to figure 5, there is a step up of the ps(t). Please explain the reason.
>
> Response: The original CMNIST dataset consists of two domains with varying spurious correlation values: 0.9 in one domain and 0.8 in the other [1]. To simulate a continuous problem, we randomly assign samples in the first domain to time indices 1-512(with interval 1)  and samples in the second domain to indices 513-1024. This mimic a situation that is collected from time 1.0 to time 1024.0 and we discretize the time index with an interval of 1. The spurious correlation only changes at time index 513, as shown in Figure 5 in the Appendix. In this case, each of the 1024 domains consists of approximately 50 samples. Based on the results presented in Figure 2 in the main part of the manuscript, REx and IRMv1 exhibit testing inaccuracies that are close to random guessing in this scenario. Similar constructions are made for 4, 8, ..., 512 domains.
>
> We re-construct the original CMNIST in [1] in this manner is trying to show that those method working well in the original setting will struggle when there are more domains and less samples per domain.
>
> [1] Martin Arjovsky, et.al., Invariant Risk Minization

---

> ### Comment · Area_Chair_PEQq · 2023-11-20
> **please add some comments about the rebuttal**
>
> Dear reviewer Zzch,
>
> this paper is still in the "borderline" region, and in order to come to a conclusion, it would be very helpful to see your comments about the other reviews and the rebuttal.
>
> Thanks again,
> AC

---

> ### Author Response · Authors · 2023-11-22
>
> Dear Reviewer Zzch:
>
> Thank you for your time and effort in reviewing our paper.
>
> We firmly believe  that  our response and revisions can fully address your concerns. We are open to discussion if you have any additional questions or concerns, and if not, we kindly ask you to reevaluate your score.
>
> Thank you again for your reviews which helped to improve our paper!
>
> Authors

---

### Author Response · Authors · 2023-11-19
**General Response**

We sincerely appreciate the time and efforts of the reviewers in providing their valuable feedback. We have incorporated the suggested modifications in our script, which are highlighted in blue.

The key points of our rebuttal can be summarized as follows:

1. Provide extensive discussion and clarifications on the following aspects

	1.1 The notation of "Continous Domain" ([link](https://openreview.net/forum?id=70IgE3tRbu&noteId=cGPxo2cazq)).

	1.2 Our theretical contributions ([link](https://openreview.net/forum?id=70IgE3tRbu&noteId=8IBddITadU)).

	1.3 The advantage of our method over InvRat and IIBNet ([link](https://openreview.net/forum?id=70IgE3tRbu&noteId=8IBddITadU)).

2. Conduct additional experiments on Heart Disease, which contains mechanically related domains in real value ([link](https://openreview.net/forum?id=70IgE3tRbu&noteId=cGPxo2cazq )).

3. Signficanlty improve the performance of CIL on Continous CMNIST ([link](https://openreview.net/forum?id=70IgE3tRbu&noteId=wburPQKZJe )).

3. Conduct experiments on additional baslines, including InvRat, EIIL, HRM and ZIN ([link](https://openreview.net/forum?id=70IgE3tRbu&noteId=8IBddITadU)).

4. Show that $O(\sqrt{n})$ is the asymptotically optimal rate of $|\mathcal{T}|$ can grow with respect to $n$ for REx to fail with a non-zero probability ([link](https://openreview.net/forum?id=70IgE3tRbu&noteId=XYRUN1oPHC)).

We eagerly await feedback from the reviewers and look forward to engaging in a fruitful discussion.

---

### Meta-Review · Area_Chair_PEQq · 2023-12-08

**Metareview:**

This paper was a borderline case. Originally, many points of criticism have been raised by the reviewers, such as the unclear precise definition of the term "continuous invariance", unclear novelty of the concept, experimental results that were not easy to interpret, etc. In the rebuttal phase, however, several of these critical comments could be addressed addressed. Although I am still not fully convinced that the conceptual novelty is very high, I think that the paper has several interesting facets and that the positive aspects outweigh the negative ones. Therefore, I recommend acceptance of this paper.

**Justification For Why Not Higher Score:**

Limited theoretical novelty compared to existing work.

**Justification For Why Not Lower Score:**

Over-all, I had the impression that this paper contains sufficiently many interesting results that could be helpful for the research community.

---

### Decision · Program_Chairs · 2024-01-16

Accept (poster)